



# S5P/TROPOMI NO$_2$ slant column retrieval: method, stability, uncertainties, and comparisons against OMI

Jos van Geffen[1], K. Folkert Boersma[1,2], Henk Eskes[1], Maarten Sneep[1], Mark ter Linden[1,3], Marina Zara[1,2], and J. Pepijn Veefkind[1,4]

[1]Royal Netherlands Meteorological Institute (KNMI), De Bilt, The Netherlands
[2]Wageningen University (WUR), Wageningen, The Netherlands
[3]Science and Technology Corporation (S[&]T), Delft, The Netherlands
[4]Delft University of Technology (TUDelft), Delft, The Netherlands

**Correspondence:** J. van Geffen (geffen@knmi.nl)

**Abstract.**

The Tropospheric Monitoring Instrument (TROPOMI), aboard the Sentinel-5 Precursor (S5P) satellite, launched on 13 Oct. 2017, provides measurements of atmospheric trace gases and of cloud and aerosol properties on an unprecedented spatial resolution of approximately $7 \times 3.5$ km$^2$ (approx. $5.5 \times 3.5$ km$^2$ as of 6 Aug. 2019), achieving near-global coverage in one day.

The retrieval of nitrogen dioxide (NO$_2$) concentrations is a 3-step procedure: slant column density (SCD) retrieval, separation of the SCD in its stratospheric and tropospheric components, and conversion of these into vertical column densities. This study focusses on the TROPOMI NO$_2$ SCD retrieval: the retrieval method used, the stability of the SCDs and the SCD uncertainties, and a comparison against OMI NO$_2$ SCDs.

The statistical uncertainty, based on the spatial variability of the SCDs over a remote Pacific Ocean sector, is $8.63$ $\mu$mol/m$^2$

for all pixels ($9.45$ $\mu$mol/m$^2$ for cloud-free pixels), which is very stable over time and some $30\%$ less than the long-term average over OMI/QA4ECV data (since the pixel size reduction TROPOMI uncertainties are $\sim 10\%$ larger). The SCD uncertainty reported by the DOAS fit is about $10\%$ larger than the statistical uncertainty, while for OMI/QA4ECV the DOAS uncertainty is some $20\%$ larger than its statistical uncertainty. Comparison of the SCDs themselves over the Pacific Ocean, averaged over one month, shows that TROPOMI is about $5\%$ higher than OMI/QA4ECV, which seems to be due mainly to the use of the so-called

intensity offset correction in OMI/QA4ECV but not in TROPOMI: turning that correction off means about $5\%$ higher SCDs. The row-to-row variation in the SCDs of TROPOMI, the "stripe amplitude", is $2.14$ $\mu$mol/m$^2$, while for OMI/QA4ECV it is $\sim 2$ ($\sim 5$) larger in 2005 (2018), still a so-called stripe correction of this non-physical across-track variation is useful for TROPOMI data. In short, TROPOMI shows a superior performance compared against OMI/QA4ECV and operates as anticipated from instrument specifications.

The TROPOMI data used in this study covers 30 April 2018 up to 31 Oct. 2019.



# 1 Introduction

Nitrogen dioxide ($NO_2$) and nitrogen oxide (NO) – together usually referred to as nitrogen oxides ($NO_x$) – enter the atmosphere due to anthropogenic and natural processes.

Over remote regions $NO_2$ is primarily located in the stratosphere, with concentrations in the range $33 - 116$ $\mu mol/m^2$ ($2 - 7 \times 10^{15}$ molec/cm$^2$) between tropics and high latitudes. Stratospheric $NO_2$ is involved in photochemical reactions with ozone and thus may affect the ozone layer, either by acting as a catalyst for ozone destruction (Crutzen, 1970; Seinfeld and Pandis, 2006; Hendrick et al., 2012) or by suppressing ozone depletion (Murphy et al., 1993).

Tropospheric $NO_2$ plays a key role in air quality issues, as it directly affects human health (WHO, 2003), with concentrations up to $500$ $\mu mol/m^2$ ($30 \times 10^{15}$ molec/cm$^2$) over polluted areas. In addition, nitrogen oxides are essential precursors for the formation of ozone in the troposphere (Sillman et al., 1990) and they influence concentrations of OH and thereby shorten the lifetime of methane (Fuglestvedt et al., 1999). $NO_2$ in itself is a minor greenhouse gas, but the indirect effects of $NO_2$ on global climate change are probably larger, with a presumed net cooling effect mostly driven by oxidation-fuelled aerosol formation (Shindell et al., 2009).

The important role of $NO_2$ in both troposphere and stratosphere requires monitoring of its concentration on a global scale, where observations from satellite instruments provide global coverage, complementary to sparse measurements by ground-based in-situ and remote sensing instruments, and measurements with balloons and aircraft. With lifetimes in the troposphere of only a few hours, the $NO_2$ stays relatively close to its source, and the observations may be used for top-down emission estimates (Schaub et al., 2007; Beirle et al., 2011; Wang et al., 2012; van der A et al., 2017).

The Tropospheric Monitoring Instrument (TROPOMI; Veefkind et al., 2012), aboard the European Space Agency (ESA) Sentinel-5 Precursor (S5P) satellite, which was launched on 13 October 2017, provides measurements of atmospheric trace gases (such as $NO_2$, $O_3$, $SO_2$, HCHO, $CH_4$, CO) and of cloud and aerosol properties on an unprecedented spatial resolution of 7.2 km (5.6 km as of 6 Aug. 2019) along-track by 3.6 km across-track at nadir, with a 2600 km wide swath, thus achieving near-global coverage in one day.

The TROPOMI $NO_2$ retrieval (van Geffen et al., 2019; Eskes et al., 2019b) uses the three step approach introduced for the OMI $NO_2$ retrieval (the DOMINO approach; Boersma et al., 2007, 2011). This approach is also applied in the QA4ECV project (Boersma et al., 2018) which provides a consistent reprocessing for the $NO_2$ retrieval from measurement by OMI aboard EOS-Aura (Levelt et al., 2006, 2018), GOME-2 aboard MetOp-A (Munro et al., 2006, 2016), SCIAMACHY aboard Envisat (Bovensmann et al., 1999), and GOME aboard ERS-2 (Burrows et al., 1999).

The first step is an $NO_2$ slant column density (SCD) retrieval using a Differential Optical Absorption Spectroscopy (DOAS) technique, which provides the total amount of $NO_2$ along the effective light path from sun through atmosphere to satellite. Next, $NO_2$ vertical profile information from a chemistry transport model / data assimilation (CTM/DA) system that assimilates the satellite observations is used to separate the stratospheric and tropospheric components of the total SCD. And finally these SCD components are converted to $NO_2$ vertical stratospheric and tropospheric column densities using appropriate air-mass factors (AMFs).



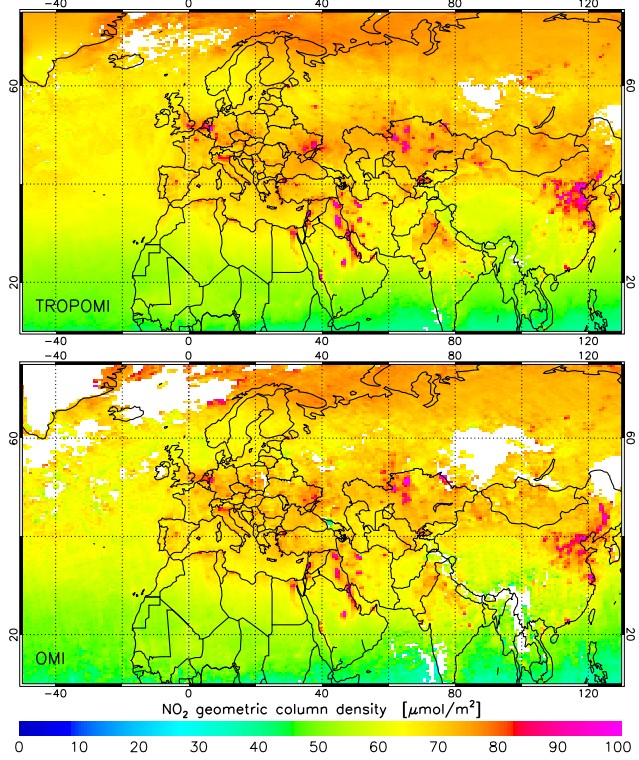

**Figure 1.** NO$_2$ geometric column density (GCD, defined in Sect. 4) from TROPOMI (top panel) and OMI/QA4ECV (bottom panel) averaged over 20–26 July 2019 on a common longitude $\times$ latitude grid of $0.8° \times 0.4°$. The OMI data is filtered for the row anomaly (Sect. 2.2.2); no other filtering is applied.

This paper focusses on the first step, the TROPOMI NO$_2$ SCD retrieval: it provides details of the retrieval method (Sect. 3), analyses the stability and uncertainties of the SCD retrieval (Sect. 4), and discusses some further issues related to the NO$_2$ SCD retrieval (Sect. 5). The TROPOMI data used in this study covers the period 30 April 2018 (which is the start of the operational (E2) phase) up to 31 Oct. 2019.

5 OMI NO$_2$ slant column data from QA4ECV (Boersma et al., 2018) can be used for comparisons (Sect. 4), because OMI and TROPOMI provide observations at almost the same local time. The example in Fig. 1 shows that both instruments capture the larger NO$_2$ hotspots equally well, but that OMI misses some smaller hotspots and that its measurements are more noisy than TROPOMI's because the latter has a higher spatial resolution and a better signal-to-noise ratio.

 TROPOMI level-2 data is reported in SI units, which for NO$_2$ means in mol/m$^2$; for convenience of the reader this paper

10 uses the SI units and in most instances also provides numbers in the more commonly used unit of molec/cm$^2$; the conversion factor between the two is $6.02214 \times 10^{19}$.



## 2 Satellite data sources and data selection

### 2.1 TROPOMI aboard Sentinel-5 Precursor

#### 2.1.1 TROPOMI instrument

The Tropospheric Monitoring Instrument (TROPOMI; Veefkind et al., 2012) is a nadir-viewing spectrometer aboard ESA's
Sentinel-5 Precursor (S5P) spacecraft, which was launched in October 2017. From an ascending sun-synchronous polar orbit,
with an equator crossing at about 13:40 local time, TROPOMI provides measurements in four channels (UV, Visible, NIR and
SWIR) of various trace gas concentrations, as well as cloud and aerosol properties. In the visible channel ($400-496$ nm), used
for the $NO_2$ retrieval, the spectral resolution and sampling are $0.54$ nm and $0.20$ nm, with a signal-to-noise ratio of around
1500. Radiance measurements are taken along the dayside of the Earth; once every 15 orbits a small part of the dayside orbit
near the north pole is used to measure the solar irradiance.

Individual ground pixels are 7.2 km (5.6 km as of 6 Aug. 2019), with integration time 1.08 s (0.84 s), in the along-track and
3.6 km in the across-track direction at the middle of the swath. There are 450 ground pixels (rows) across-track and their size
remains more or less constant towards the edges of the swath (the largest pixels are $\sim 14$ km wide). The full swath width is
about 2600 km and with that TROPOMI achieves global coverage each day, except for narrow strips between orbits of about
$0.5°$ wide at the equator. Along-track there are 3245 or 3246 scanlines (4172 or 4173 after the along-track pixel size reduction)
in regular radiance orbits, leading to about 1.46 (1.88) million ground pixels per orbit; for orbits with irradiance measurements
there are about $10\%$ less scanlines. Approximately $15\%$ of the ground pixels are not processed due to the limit on the solar
zenith angle ($\theta_0 \leq 88°$) in the processing.

#### 2.1.2 TROPOMI observations used in this study

The TROPOMI $NO_2$ data retrieval is described in the product ATBD (van Geffen et al., 2019); see also the Product User
Manual (Eskes et al., 2019a) and the Product ReadMe File (Eskes and Eichmann, 2019) for usage of the data and the data
product versions.

To investigate the stability and uncertainties of the TROPOMI $NO_2$ SCDs, orbits over the Pacific Ocean, i.e. away from
sources of $NO_2$, are used: for each day the first available orbit with satellite (nadir viewing) equator crossings west of about
$-135°$. Such an orbit is missing on a few days and these days are thus skipped.

The TROPOMI data used in this study covers the period 30 April 2018 (which is the start of the operational (E2) phase)
up to 31 Oct. 2019. Off-line (re)processed data of versions 1.2.x and 1.3.x are used; these versions do not differ in the SCD
retrieval part of the processing and are based on level-1b version 1.0.0 spectra (Babić et al., 2017). Near-real time (NRT) data
are not considered here; validation of both the off-line and NRT data has shown that results of these processing chains do not
differ significantly (Lambert et al., 2019).



### 2.2 OMI aboard EOS-Aura

#### 2.2.1 OMI instrument

The Ozone Monitoring Instrument (OMI; Levelt et al., 2006) is a nadir-viewing spectrometer aboard NASA's EOS-Aura spacecraft, which was launched in July 2004. From an ascending sun-synchronous polar orbit, with an equator crossing at about

13:30 local time, OMI provides measurements in three channels (two UV and one Visible) of various trace gas concentrations, as well as cloud and aerosol properties. In the visible channel ($349-504$ nm), used for the $NO_2$ retrieval, the spectral resolution and sampling are $0.63$ nm and $0.21$ nm, with a signal-to-noise ratio of around $500$. Radiance measurements are taken along the dayside of the Earth; once every 15 orbits a small part of the dayside orbit near the north pole is used to measure the solar irradiance.

Individual ground pixels are 13 km, with integration time 2 s, in the along-track and 24 km in the across-track direction at the middle of the swath. There are 60 ground pixels (rows) across-track and their size increases towards the edges of the swath to $\sim 150$ km. The full swath width is about 2600 km and with that OMI achieves global coverage each day. Along-track there are 1643 or 1644 scanlines in regular radiance orbits, leading to just under 100,000 ground pixels per orbit; for orbits with irradiance measurements there are about 10% less scanlines.

#### 2.2.2 OMI observations used in this study

Comparisons of the magnitude of the $NO_2$ SCDs of TROPOMI and OMI is done using OMI orbits from 2018-2019 as processed within the framework of the QA4ECV project (Boersma et al., 2018). Since June 2007 a part of the OMI detector suffers from a so-called row anomaly, which appears as a signal suppression in the level-1b radiance data at all wavelengths (Schenkeveld et al., 2017), leading e.g. to large uncertainties on the $NO_2$ SCDs in the affected rows $22-53$ (0-based). Com-

parisons of the $NO_2$ SCD uncertainties (Sect. 4.1) are also made with OMI Pacific Ocean orbits from 2005-2006, the first year after launch, before the row anomaly occured. Note that the OMI degradation over the past 15 years is small: the SCD statistical uncertainties and SCD error estimates have increased by about 1% and 2% per year, respectively (Zara et al., 2018).

   TROPOMI and OMI measure at about the same local time (the equator crossing local time differs by about 10 min.) but since TROPOMI travels at about 830 km and OMI at about 715 km altitude, TROPOMI orbits take a little longer than OMI's:

if TROPOMI has completed one orbit, OMI has covered $\sim 1.03$ orbits. This means that if a given two orbits exactly overlap, then 19 orbits later TROPOMI's equator crossing longitude lies in between the equator crossing longitudes of two OMI orbits, i.e. a longitudinal mismatch of about $12.5°$. The difference in orbit overlap plays a role when comparing results from individual orbits (as done in Sect. 4.1) but is not relevant in case gridded averaged data are used (as done in Fig. 1 and Sect. 4.4).

### 2.3 Latitudinal range for uncertainty studies

To investigate the stability and uncertainties of the $NO_2$ SCD retrieval the "Tropical Latitude" (TL hereafter) range is defined as follows: all scanlines within a $30°$ satellite latitude range that moves along with the seasons, in an attempt to filter out





**Table 1.** Specifics for the $NO_2$ slant column retrieval of TROPOMI and OMI/QA4ECV. The reference spectra (second group of entries) have all been convolved with the row-dependent instrument spectral response function (ISRF, or: slit function).

|  | TROPOMI | OMI/QA4ECV | remark / reference / data source |
|---|---|---|---|
| type of DOAS fit | intensity fit |  | van Geffen et al. (2015); van Geffen et al. (2019) |
|  |  | optical density fit | Danckaert et al. (2017); Boersma et al. (2018) |
| $\chi^2$ minimisation method | Optimal Estimation |  | with Gauss-Newton; Rodgers (2000) |
|  |  | Levenberg-Marquardt | Press et al. (1997, Ch. 15) |
| reference spectrum in $R_{meas}$ | daily $E_0$ [a] |  | measured once per 15 orbits, i.e. every $\sim$25h:22m |
|  |  | 2005-average $E_0$ | average of OMI irradiance measurements in 2005 |
| level-1b uncertainty in $\chi^2$ | included | not included | — |
| wavelength range | $405 - 465$ nm | $405 - 465$ nm | — |
| DOAS polynomial degree | $n_p = 5$ | $n_p = 4$ | number of coefficients is $n_p + 1$ |
| intensity offset correction | not included | constant | — |
| solar reference spectrum | $E_{ref}$ | $E_{ref}$ | UV/Vis channel: Dobber et al. (2008) |
| $NO_2$ reference spectrum | $\sigma_{NO_2}$ at 220 K | $\sigma_{NO_2}$ at 220 K | Vandaele et al. (1998) |
| ozone reference spectrum | $\sigma_{O_3}$ at 223 K | $\sigma_{O_3}$ at 243 K | Serdyuchenko et al. (2014) |
| $O_2$-$O_2$ reference spectrum | $\sigma_{O_2\text{-}O_2}$ at 293 K | $\sigma_{O_2\text{-}O_2}$ at 293 K | Thalman and Volkamer (2013) |
| water vapour reference spectrum | $\sigma_{H_2O_{vap}}$ at 293 K | $\sigma_{H_2O_{vap}}$ at 293 K | HITRAN 2012: Rothman et al. (2013) |
| liquid water reference spectrum | $\sigma_{H_2O_{liq}}$ | $\sigma_{H_2O_{liq}}$ | Pope and Fry (1997) |
| Ring reference spectrum | $I_{ring}$ | $\sigma_{ring}$ | derived following Chance and Spurr (1997) |
| level-2 off-line data version | v1.2.x & v1.3.x |  | https://s5phub.copernicus.eu/ |
|  |  | v1.1 | http://www.qa4ecv.eu/ |
| level-1b off-line data version | v1.0.0 |  | https://s5phub.copernicus.eu/ |
|  |  | coll. 3 | https://disc.gsfc.nasa.gov/ |

[a]) Off-line (re)processing uses $E_0$ measured nearest in time to $I$, except the period mid-Oct. 2018 to mid-March 2019, when the most recent $E_0$ w.r.t. $I$ was used due to an issue with the processor; the version-2 reprocessing will use the nearest $E_0$ for all orbits.

seasonality in the $NO_2$ columns: on 1 January the TL range covers $[-30° : 0°]$ for the nadir viewing rows, while half a year later it is $[0 : +30°]$. The TL range is also used for the across-track "de-striping" of the SCDs discussed in Sect. 4.3. For TROPOMI (OMI) data the TL range contains about 475 (250) scanlines; after the along-track pixel size reduction of TROPOMI there are about 610 scanlines in the TL range.



## 3 NO$_2$ slant column retrieval

Though this paper discusses method and results of the TROPOMO NO$_2$ slant column retrieval (Sect. 3.2), it is important to also discuss the retrieval method used for OMI data within the QA4ECV (Sect. 3.3) and OMNO2A (Sect. 3.4) approaches, because differences in results (Sect. 4) turn out to be mainly related to retrieval method details.

### 3.1 DOAS technique

The NO$_2$ slant column density (SCD) retrieval is performed using a Differential Optical Absorption Spectroscopy (DOAS) technique (Platt, 1994; Platt and Stutz, 2008), which provides the amount of NO$_2$ along the effective light path, from sun through atmosphere to satellite. This technique attempts to model the reflectance spectrum $R_{\mathrm{meas}}(\lambda)$ observed by the satellite instrument:

$$R_{\mathrm{meas}}(\lambda) = \frac{\pi\, I(\lambda)}{\mu_0\, E_0(\lambda)} \tag{1}$$

with $I(\lambda)$ the radiance at the top of the atmosphere, $E_0(\lambda)$ the extraterrestrial solar irradiance measured by the same instrument, and $\mu_0 = \cos(\theta_0)$ the cosine of the solar zenith angle; given that the processing is limited to ground pixels measured at $\theta_0 \leq 88°$, the division by $\mu_0$ in Eq. (1) will not cause problems. Note that both $I$ and $E_0$ also depend on viewing geometry, but those arguments are left out for brevity.

The modelled reflectance, $R_{\mathrm{mod}}(\lambda)$, is determined from reference spectra of a number of species known to absorb in the wavelength window used for the SCD retrieval, as well as a correction for scattering and absorption by rotational Raman scattering, the so-called "Ring effect" (see Grainger and Ring, 1962; Chance and Spurr, 1997), while a polynomial $P(\lambda) = \sum a_m \lambda^m$ $(m = 0, 1, \ldots, n_p)$ is used to account for spectrally smooth structures resulting from molecular (single and multiple) scattering and absorption, aerosol scattering and absorption, and surface albedo effects.

The precise formulation of $R_{\mathrm{mod}}(\lambda)$ and the method used to minimise the difference between the modelled and measured reflectance differs slightly between the TROPOMI and OMI retrievals. Details of these DOAS approaches are listed in Table 1. (The difference in the degree of the DOAS polynomial is not relevant: $n_p = 4$ and $n_p = 5$ give pratically the same results; for TROPOMI $n_p = 5$ is chosen following the traditional setting in the OMNO2A processing (Sect. 3.4) of OMI data.)

### 3.2 TROPOMI intensity fit retrieval

In the TROPOMI NO$_2$ processor (van Geffen et al., 2019) $R_{\mathrm{mod}}(\lambda)$ is formulated in an intensity fit (IF hereafter) approach:

$$R_{\mathrm{mod}}(\lambda) = P(\lambda) \cdot \exp\left[ -\sum_{k=1}^{n_k} \sigma_k(\lambda) \cdot N_{\mathrm{s},k} \right] \cdot \left( 1 + C_{\mathrm{ring}} \frac{I_{\mathrm{ring}}(\lambda)}{E_0(\lambda)} \right). \tag{2}$$

with $\sigma_k(\lambda)$ the absolute cross section and $N_{\mathrm{s},k}$ the slant column amount of molecule $k = 1, \ldots, n_k$ taken into account in the fit: NO$_2$, ozone, water vapour, liquid water, and the O$_2$-O$_2$ collision complex. The physical model accounts for inelastic Raman scattering of incoming sunlight by N$_2$ and O$_2$ molecules that leads to filling-in of the Fraunhofer lines in the radiance spectrum, i.e. the Ring effect. In Eq. (2), $C_{\mathrm{ring}}$ is the Ring fit coefficient and $I_{\mathrm{ring}}(\lambda)/E_0(\lambda)$ the sun-normalised synthetic Ring





spectrum, with $E_0(\lambda)$ the measured irradiance. The term between parentheses in Eq. (2) describes both the contribution of the direct differential absorption (i.e. the 1), and the modification of these differential structures by inelastic scattering (the $+C_{\text{ring}} I_{\text{ring}}(\lambda)/E_0(\lambda)$ term) to the reflectance spectrum.

The IF minimises the chi-squared merit function:

$$\chi^2 = \sum_{i=1}^{n_\lambda} \left( \frac{R_{\text{meas}}(\lambda_i) - R_{\text{mod}}(\lambda_i)}{\Delta R_{\text{meas}}(\lambda_i)} \right)^2 \qquad (3)$$

with $n_\lambda$ the number of wavelengths (spectral pixels) in the fit window ($405 - 465$ nm) and $\Delta R_{\text{meas}}(\lambda_i)$ the uncertainty on the measured reflectance, which depends on the precision of the radiance and irradiance measurements as given in the level-1b product, i.e. on the signal-to-noise ratio (SNR) of the measurements. Radiance spectral pixels flagged in the level-1b data as bad or as suffering from saturation effects are filtered out before any further processing step.

In the final data product ground pixels are flagged when the slant column retrieval uncertainty $\Delta N_s > 33$ $\mu$mol/m$^2$ ($2 \times 10^{15}$ molec/cm$^2$). SCD error values this large occur rarely: usually $< 0.1\%$ ($< 0.2\%$) of the pixels per orbit with original (smaller) size ground pixels (note that the ground pixel size reduction leads to about $28\%$ more ground pixels per orbit, so that with the size reduction the number of succesfully retrieved pixels increases significantly).

The magnitude of $\chi^2$ is a measure for how good the fit is. Another measure for the goodness of the fit is the so-called
root-mean-square (RMS) error:

$$R_{\text{RMS}} = \sqrt{\frac{1}{n_\lambda} \sum_{i=1}^{n_\lambda} \left( R_{\text{meas}}(\lambda_i) - R_{\text{mod}}(\lambda_i) \right)^2} \qquad (4)$$

where the difference $R_{\text{res}}(\lambda) = R_{\text{meas}}(\lambda) - R_{\text{mod}}(\lambda)$ is usually referred to as the residual of the fit.

In the TROPOMI processor $\chi^2$ is minimised using an Optimal Estimation (OE; based on Rodgers, 2000) routine, with suitable a-priori values of the fit parameters and a-priori errors set very large, so as not to limit the solution of the fit (for
example, the NO$_2$ SCD a-priori error is set at $1.0 \times 10^{-2}$ mol/m$^2 = 6 \times 10^{17}$ molec/cm$^2$), while for numerical stability reasons a pre-whitening of the data is performed. Estimated slant column and fitting coefficient uncertainties are obtained from the diagonal of the covariance matrix of the standard errors, while the off-diagonal elements represent the correlation between the fit parameters[1]. The SCD error estimates are scaled with the square-root of the normalised $\chi^2$, where $\chi^2$ is normalised by $(n_\lambda - D)$, with $D$ the degrees of freedom of the fit, which is almost equal to the number of fit parameters:
$\Delta N_s = \Delta N_s^{\text{OE}} \cdot \sqrt{\chi^2/(n_\lambda - D)}$, with $\Delta N_s^{\text{OE}}$ the SCD error reported by the OE routine. The NO$_2$ output data product provides $\Delta N_s$, $\chi^2$, $n_\lambda$, $D$, and RMS error.

### 3.2.1 TROPOMI wavelength calibration

Before forming the reflectance of Eq. (1) both $I(\lambda)$ and $E_0(\lambda)$ are calibrated, after which the calibrated $E_0(\lambda_{\text{cal}})$ is interpolated, using information from a high-resolution reference spectrum ($E_{\text{ref}}$; see Table 1), to the calibrated $I(\lambda_{\text{cal}})$, which serves as the

---

[1] The correlation coefficients, however, are not available in the current TROPOMI data product.



common grid for the reflectance. In the TROPOMI processor these steps are performed prior to the DOAS fit (van Geffen et al., 2019).

A wavelength calibration essentially replaces the nominal wavelength $\lambda_{\mathrm{nom}}$ that comes along with the level-1b spectra by a calibrated version:

$$\lambda_{\mathrm{cal}} = \lambda_{\mathrm{nom}} + w_s + w_q(\lambda_{\mathrm{nom}} - \lambda_0) \tag{5}$$

where $w_s$ represents a wavelength shift and $w_q$ a wavelength stretch ($w_q > 0$) or squeeze ($w_q < 0$), with $w_q$ defined w.r.t. the central wavelength of the fit window $\lambda_0$. Each radiance ground pixel and each irradiance row has its own wavelength grid and calibration results. In the TROPOMI processor fitting $w_q$ is turned off; see below for a short discussion of this.

The wavelength calibration is performed over the full $NO_2$ fit window ($405-465$ nm), using a high-resolution solar reference spectrum ($E_{\mathrm{ref}}$, pre-convolved with the TROPOMI ISRF; see Table 1) and the OE routine also used for solving the DOAS equation. For the $I(\lambda)$ calibration a 2nd order polynomial as well as a term representing the Ring effect are included: the model function used for the radiance wavelength calibration is a modified version of Eq. (2); for the $E_0(\lambda)$ calibration the Ring term is obviously excluded.

Fig. 2a shows the wavelength shifts $w_s$ for an orbit on 1 July 2018 of the irradiance (red) and radiance (blue) as function of across-track ground pixel (row), where the radiance shift of each row is an along-track average over the Tropical Latitude (TL) range defined in Sect. 2.3. Due to only partial instrument slit illumination at the outer two rows, 0 and 449, $w_s$ shows markedly different values for these rows. To avoid these peaks from overshadowing the effects discussed below, the outer two rows are skipped from the following analysis.

The broad across-track shape and the average value of $w_s$ visible in Fig. 2a are not important, as they result from the choice of the nominal grid of the level-1b data. The change in time of the average $\overline{w_s}$ and of the row-to-row variation in $w_s$, however, give an idea of the stability of the level-1b data and hence of the instrument. Fig. 2b shows the temporal change of $\overline{w_s}$. There seems to be a small long-term oscillation in this, with an amplitude of about $0.0015$ nm and $0.0020$ nm for radiance and irradiance, respectively, which looks like to be a seasonal effect. A similar seasonal variation, though larger in magnitude, is seen in the OMI wavelength calibration data, where it may be attributed to a seasonal cycle in the temperature of the optical bench (S. Marchenko, pers. comm., 2019). Unlike in OMI, that temperature in TROPOMI is actively stabilised and therefore shows changes with a much smaller amplitude than in OMI (Q. Kleipool, pers. comm., 2019), resulting in a lower seasonal amplitude in $\overline{w_s}$.

The dominant term in the overall magnitude of the radiance is the inhomogeneous illumination of the instrument slit as a result of the presence of clouds, which may show up as differences in $w_s$. The magnitude of the day-to-day variation in the average is much smaller than this long-term oscillation. The row-to-row variation in the shift, visible in Fig. 2a, is small and the evolution of that across-track variation shows a slow increase over time (not shown), probably related to degradation of the instrument (E. Loots, pers. comm., 2019).

With the forthcoming update of the level-1b data to v2.0.0 the nominal UV-Vis wavelength grids of both irradiance and radiance are adjusted by $0.027$ nm, for all rows and all days. As a result of this the average $\overline{w_s}$ will be reduced by that amount, but

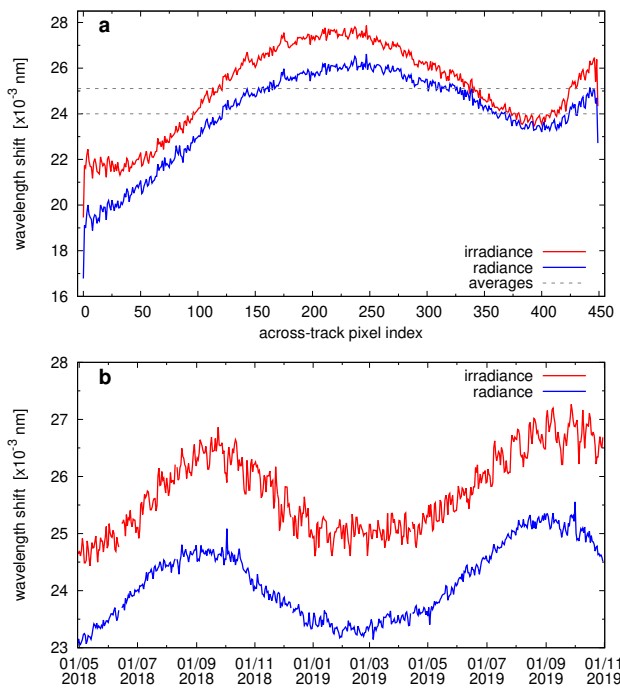

**Figure 2.** Wavelength calibration shifts $w_s$ for the NO$_2$ fit window $(405 - 465$ nm$)$ of the TROPOMI irradiance (red) and radiance (blue), where the latter is an average over the Tropical Latitude (TL) range. **a)** Shifts for 1 July 2018 (radiance orbit 03711, with irradiance from orbit 03718) as function of the across-track ground pixel index; the dashed horizontal lines are the across-track averages, with the exception of the outer rows. **b)** Time evolution of the across-track average shifts.

the across-track and in-time variations will remain the same. Level-1b v2.0.0 will contain an improved degradation correction (Rozemeijer and Kleipool, 2019), probably reducing the slow increase over time of the across-track variation mentioned above. All in all, the wavelength calibration results show that TROPOMI is a rather stable instrument, but futher monitoring of the wavelength shifts seems worthwhile.

5     Turning on the stretch fit parameter in the radiance calibration for orbit 03711 leads to a small stretch of $0.2 - 5 \times 10^{-4}$, depending on latitude, with an associated error estimate of $3 - 6 \times 10^{-4}$ (averaging over $30°$ latitude ranges with varying central latitudes): the stretch found is smaller than its error for most latitudes. At the same time the radiance wavelength shift, the NO$_2$ SCD and SCD error, and the RMS error of the DOAS fit, change on average by less than $1\%$, with a standard deviation comparable to that change or larger. In other words: including the stretch fit parameter in the radiance calibration does not

10    significantly alter the retrieval results, and hence the $w_q$ fit parameter will remain turned off.





### 3.3 OMI/QA4ECV optical density fit retrieval

The OMI data are processed in the QA4ECV framework with the QDOAS software (Danckaert et al., 2017), wherein $R_{\mathrm{mod}}(\lambda)$ is formulated in an optical density fit (ODF hereafter) approach:

$$\ln\left[R_{\mathrm{mod}}(\lambda)\right] = P(\lambda) - \sum_{k=1}^{n_k}\sigma_k(\lambda)\cdot N_{\mathrm{s},k} \; - \; \sigma_{\mathrm{ring}}(\lambda)\cdot C_{\mathrm{ring}} \;, \tag{6}$$

with $\sigma_{\mathrm{ring}}(\lambda)$ the differential (pseudo-absorption) reference spectrum of the Ring effect and $C_{\mathrm{ring}}$ its fitting coefficient, where $\sigma_{\mathrm{ring}}(\lambda)$ equals $I_{\mathrm{ring}}(\lambda)/E_{\mathrm{ref}}(\lambda)$ minus a 2nd order polynomial, with $E_{\mathrm{ref}}$ a (constant) solar reference spectrum (which is different from the measured solar spectrum $E_0(\lambda)$ used in Eq. (2)). Note that except for the way the Ring effect is treated, the IF and ODF modelled reflectances are to first order the same; see App. A for a discussion of this difference.

The ODF minimises the merit function (cf. Eq. (3)):

$$\chi^2_{\mathrm{ODF}} = \sum_{i=1}^{n_\lambda}\left(\ln\left[R_{\mathrm{meas}}(\lambda_i)\right] - \ln\left[R_{\mathrm{mod}}(\lambda_i)\right]\right)^2 \tag{7}$$

without weighting with the level-1b uncertainty estimate $\Delta R_{\mathrm{meas}}$, though QDOAS has the option to include the weighting. To minimise $\chi^2_{\mathrm{ODF}}$, QDOAS uses a Levenberg-Marquardt non-linear least-squares fitting procedure (Press et al., 1997), which also provides an estimate of the uncertainties in the fit parameters.

In the ODF formulation the RMS error is defined as:

$$R^{\mathrm{ODF}}_{\mathrm{RMS}} = \sqrt{\frac{1}{n_\lambda}\sum_{i=1}^{n_\lambda}\left(\ln\left[R_{\mathrm{meas}}(\lambda_i)\right] - \ln\left[R_{\mathrm{mod}}(\lambda_i)\right]\right)^2} \tag{8}$$

which is different from the $R_{\mathrm{RMS}}$ of the intensity fit as given in Eq. (4); see App. B for a relationship between the two.

Like many other DOAS applications, the OMI/QA4ECV processing includes a correction for an intensity offset in the radiance:

$$R_{\mathrm{meas}}(\lambda) = \frac{\pi I(\lambda + P_{\mathrm{off}}(\lambda)\cdot S_{\mathrm{off}})}{\mu_0 \, E_0(\lambda)} \tag{9}$$

with $P_{\mathrm{off}}(\lambda)$ a low-order polynomial (in OMI/QA4ECV a constant) and $S_{\mathrm{off}}$ some suitable scaling factor (QDOAS computes this dynamically from some average of the measured solar spectrum $E_0(\lambda)$ in the DOAS fit window). Sect. 5.1 discusses the possible origin and implication of this correction term.

QDOAS also has the option to be run in intensity fit mode, in which case the modelled reflectance includes the Ring effect as a pseudo-absorber like it does in the optical density fit mode Eq. (6), rather then as the non-linear term like in Eq. (2).

### 3.3.1 OMI/QA4ECV wavelength calibration

In QDOAS (Danckaert et al., 2017) the wavelength calibration of $E_0(\lambda)$ is performed prior to the DOAS fit, based on a high-resolution solar reference spectrum ($E_{\mathrm{ref}}$; see Table 1). The calibration of $I(\lambda)$ is part of the DOAS fit: the shift, $w_s$, and stretch,





$w_q$, are fitted along with the SCDs, with the calibrated $E_0(\lambda_{\mathrm{cal}})$ wavelength grid as the common grid for the reflectance. For OMI/QA4ECV both a shift and stretch are fitted; cf. Eq. (5). When processing TROPOMI data with QDOAS, only shifts are fitted, as is the case for the regular TROPOMI processing.

Processing the TROPOMI orbit for which the wavelength shifts are shown in Fig. 2a with QDOAS leads to almost identical

wavelength shifts: the irradiance and TL average radiance shifts differ by $0.25 \pm 0.10 \times 10^{-3}$ nm and $0.65 \pm 0.08 \times 10^{-3}$ nm, respectively (the TROPOMI spectral sampling is 0.20 nm; Sect. 2.1.1). Consequently, the difference in radiance wavelength calibration between TROPOMI and QDOAS will not affect comparisons of the retrieval results noticeably.

### 3.4 OMI/OMNO2A intensity fit retrieval

The official OMI $NO_2$ SCD data processing, running at NASA, is called OMNO2A. OMNO2A v1.2.x delivers the SCD data

for the DOMINO v2 $NO_2$ VCD processing (results of which are released via http://www.temis.nl/airpollution/no2.html). van Geffen et al. (2015) investigated a number of improvements intended for OMNO2A v2.0, which has not yet been implemented, but the SCD retrieval of OMNO2A v2.0 can be run locally at KNMI for testing and comparisons.

OMNO2A v2.0 uses the intensity fit approach with the modelled reflectance formulated in the same manner as TROPOMI, viz. Eq. (2) and the settings listed for TROPOMI in Table 1, with the exception that $\chi^2$ is minimised using a Levenberg-

Marquardt solver and wavelength calibration is performed over part of the $NO_2$ fit window (van Geffen et al., 2015). KNMI has a local tool to convert the OMI level-1b data into the TROPOMI level-1b format, enabling direct comparisons between the two processors.

### 4 $NO_2$ slant column retrieval evaluation

This section discusses the $NO_2$ SCD retrieval results of selected TROPOMI orbits in comparison with OMI orbits and addi-

tional retrieval results using QDOAS (Danckaert et al. (2017); version r1771, dd. 20 March 2018 is used here).

The SCD depends strongly on the along-track and across-track variation in solar zenith angle ($\theta_0$) and viewing zenith angle ($\theta$). To make evaluations and comparisons easier, the SCD is divided by the geometric AMF, defnded as $M_{\mathrm{geo}} = 1/\cos(\theta_0) + 1/\cos(\theta)$, which is a simple but realistic approximation for the air-mass factor for stratospheric $NO_2$. The resulting $NO_2$ total column may be called the geometric column density (GCD), to distinguish it from the total, tropospheric and stratospheric

VCDs, which are determined using AMFs based on $NO_2$ profile information coming from the CTM/DA model (see Sect. 1).

### 4.1 GCD and SCD error comparison for one orbit

Fig. 3 provides comparisons of the GCD (left column) and SCD error estimate from the DOAS fit (right column), averaged over the TL range for the Pacific Ocean orbits of TROPOMI and OMI on 1 July 2018. In view of the OMI row anomaly, the corresponding OMI orbit of 1 July 2005 is shown as well, noting that the $NO_2$ concentrations in 2005 are likely to be different

from those in 2018.





**Figure 3.** $NO_2$ geometric column density (GCD, defined in Sect. 4; left column) and slant column density (SCD) error estimate from the DOAS fit (right column) averaged over the TL range as function of the across-track viewing zenith angle ($\theta$) of Pacific Ocean orbits of TROPOMI and OMI on 1 July 2018 and of OMI on 1 July 2005. **a,d)** Regular TROPOMI processing of TROPOMI compared against OMI/QA4ECV processing. **b,e)** Regular TROPOMI processing of TROPOMI compared against QDOAS processing with TROPOMI settings and with QA4ECV settings. **d,f)** Regular TROPOMI processing of OMI compared against OMI/QA4ECV and OMNO2A (v2) results.

### 4.1.1 Geometric column density (GCD)

In Fig. 3a the GCD results of the regular TROPOMI processing are compared against the OMI/QA4ECV processing. The TROPOMI and OMI GCD of 1 July 2018 compare well in magnitude, in as far as such a comparison is possible in view of the





**Table 2.** NO$_2$ geometric column density (GCD), slant column density (SCD) error and RMS error from the DOAS fit averaged over the TL range and the central 150 detector rows of TROPOMI Pacific orbit 03711 of 1 July 2018 retrieved with QDOAS using different settings. For comparison, the regular v1.2.2 TROPOMI results (used in this study) and a local reprocessing using the forthcoming v2.0.0 are also listed. Given the difference in RMS error definitions, their values from QDOAS and TROPOMI retrievals cannot be compared directly (Sect. 3.3).

| processor | case | DOAS type | int. off. correction | GCD [$\mu$mol/m$^2$] | SCD error [$\mu$mol/m$^2$] | RMS error [$10^{-4}$] | remark |
|---|---|---|---|---|---|---|---|
| QDOAS | 1 | ODF | no | $45.93 \pm 0.99$ | $9.39 \pm 0.25$ | $8.10 \pm 0.21$ | |
| | 2 | ODF | yes | $43.51 \pm 0.79$ | $8.57 \pm 0.29$ | $7.36 \pm 0.24$ | QA4ECV config. |
| | 3 | IF | no | $46.45 \pm 1.03$ | $9.31 \pm 0.26$ | $8.82 \pm 0.21$ | TROPOMI config. |
| | 4 | IF | yes | $44.22 \pm 0.85$ | $8.68 \pm 0.29$ | $8.10 \pm 0.23$ | |
| TROPOMI | a | IF | no | $46.34 \pm 0.95$ | $8.93 \pm 0.22$ | $2.22 \pm 0.35$ | v1.2.2 |
| | b | IF | no | $46.94 \pm 1.00$ | $9.18 \pm 0.21$ | $2.21 \pm 0.35$ | v2.0.0 [a] |
| | c | IF | yes | $45.30 \pm 0.87$ | $8.65 \pm 0.19$ | $2.08 \pm 0.35$ | v2.0.0 [a] |

[a]) With respect to v1.2.2, v2.0.0 entails two small bug fixes and spike removal (Sect. 4.1.3).

large row-to-row variation in the OMI data and the row-anomaly: averaged over the viewing zenith angle range $\theta = [-55° : -10°]$ TROPOMI's GCD is about $3\%$ higher than OMI's. Near the western (left) edge of the swath, TROPOMI seems to report lower NO$_2$ values than OMI, which might be related to the fact that nadir of the OMI orbit lies $9°$ east of TROPOMI nadir. The OMI GCD of 1 July 2005 clearly shows less row-to-row variation than the OMI 2018 data, but more than the TROPOMI data (cf. Sect. 4.3).

In Fig. 3b the regular TROPOMI results are compared against a processing of the TROPOMI level-1b data with QDOAS, using settings as close as possible to those of the TROPOMI processor and settings used for QA4ECV (viz. Table 1). When using TROPOMI settings the QDOAS results match very closely to those of the regular TROPOMI processing: averaged over the central 150 (of the 450) detector rows the difference is about $0.2\%$. The QDOAS QA4ECV settings are different from

the TROPOMI settings at three points (type of DOAS fit, use of level-1b uncertainly in $\chi^2$ minimisation and intensity offset correction), as a result of which the GCDs (and thus the SCDs) are lower by about $6.1\%$ for this orbit. Sect. 4.2 discusses the effect of the QDOAS settings somewhat further.

   In Fig. 3c the OMI results of the regular QA4ECV processing are compared against a processing of the OMI level-1b data with the OMNO2A and TROPOMI SCD processors for the OMI orbit of 2005 in Fig. 3a, in order to investigate the impact

of retrieval method details. As with the TROPOMI data in Fig. 3b, the QA4ECV settings clearly give the lowest GCD results: averaged over the central 20 (of the 60) detector rows, the QA4ECV GCD is lower than the OMNO2A processor GCD by about $3.7\%$ and lower than the TROPOMI processor GCD by about $7.0\%$.


### 4.1.2 Slant column density (SCD) error

In the case of TROPOMI, on-board across-track binning of measurements takes place: for the outer 22 (20) rows at the left (right) edge of the swath, the binning factor is 1, while for the other rows 2 detector pixels are combined, in order to keep the across-track ground pixel width more or less constant. As a result of this, the outer rows have a larger spectral uncertainty,

which is reflected in a larger SCD error. The increased SCD error visible in the TROPOMI data of Fig. 3d-e around $\theta \approx +20°$ is related to the presence of saturation effects above bright clouds along this particular orbit.

Fig. 3d-f shows that the SCD error estimate for TROPOMI data is considerably lower than the estimates for OMI/QA4ECV data. Given that the TROPOMI and OMI retrievals are performed with different methods, a direct comparison between SCD error is only tentative; an independent method to compare SCD uncertainties is discussed in Sect. 4.6. Averaged over $\theta =$

$[-55° : -10°]$, i.e. away from the row-anomaly, TROPOMI's SCD error is about $40\%$ ($30\%$) lower than OMI's 2018 (2005) data.

The reason why the OMI SCD error in 2018 is higher than in 2005 (Fig. 3d) is, at least partly, related to the fact that in the OMI processing the one-year average irradiance of 2005 is used for all retrievals, and the larger the time difference between radiance and irradiance measurements, the larger the error on the reflectance and thus on the SCD error to be (cf. Sect. 4.5).

This issue has been discussed in detail by Zara et al. (2018).

Fig. 3e shows that the TROPOMI SCD error estimate compares reasonably well with the estimate provided by QDOAS, despite the differences in retrieval methods: averaged over the central 150 detector rows the difference is about $+4.2\%$ with TROPOMI settings and about $-2.0\%$ with QA4ECV settings (see also Sect. 4.2). Fig. 3f shows that in case of OMI data the SCD error is lowest for the regular QA4ECV retrieval: the TROPOMI processor report a $10.2\%$ higher and the OMNO2A

processor a $15.4\%$ higher SCD error.

### 4.1.3 Impact of NO$_2$ processor updates to v2.0.0

An update of the level-2 NO$_2$ SCD data to version 2.0.0 (planned for early 2020) entails two small bug fixes in the wavelength assignment and better treatment of saturated radiance wavelength pixels and of outliers in the residual (App. C). These improvements will have a small impact on the absolute value of the NO$_2$ SCD, SCD error and RMS error of the fit: on aver-

age $+0.5\%$, $+2.5\%$ and $-1\%$, respectively, based on a set of test orbits (see also Table 2). These changes are, however, not expected to alter the averages and temporal stability presented in this paper significantly.

TROPOMI level-1b spectra version 1.0.0 suffer from a small degradation (Rozemeijer and Kleipool, 2019) of $1-2\%$, notably in the irradiance. The update of the level-1b spectra to version 2.0.0 (planned for early 2020) will include a correction for the degradation, as well as some calibration corrections. This update will have a small impact on the absolute value of the NO$_2$

SCD, SCD error and RMS error of the fit: on average $+2\%$, $-1\%$ and $-6\%$, respectively, based on the evaluation of 12 test orbits. A reprocessing of all E2 phase data using v2.0.0 level-1b spectra and NO$_2$ v2.0.0 will probably take place sometime in 2020.



## 4.2 TROPOMI NO$_2$ SCD: different QDOAS options

As mentioned in the previous section (and visible in Fig. 3), the retrieval results depend on the details of the DOAS NO$_2$
SCD retrieval: the type of the DOAS fit (IF or ODF) and the retrieval settings used (in particular whether the intensity offset
correction is included or not).

Table 2 presents the GCD, SCD error and RMS error of the DOAS fit for four combinations of QDOAS settings when
processing TROPOMI orbit 03711, with other configuration settings as much as possible those of the TROPOMI processor
(if included, the intensity offset correction polynomial $P_{\mathrm{off}}(\lambda)$ is a constant), as well as the results from the TROPOMI NO$_2$
processor. Conclusions from these results:

– Turning on the intensity offset correction in QDOAS has quite a large impact on the results: the GCD goes down by $\sim 5\%$,
while the SCD error goes down by $\sim 8\%$.

– That turning on the intensity offset correction in QDOAS leads to a lower RMS error is logical, since an extra fit parameter
is introduced; it cannot be determined which part of the reduction of the RMS error (by $\sim 9\%$) is due to this extra fit parameter
and which part is due to a physically better fit.

– In IF mode QDOAS retrieves slightly larger GCDs ($\sim 1\%$) and slightly lower SCD errors ($\sim 1\%$), showing that the precise
fit method itself does not affect the fit results much.

– That the RMS error in QDOAS IF mode is $\sim 9\%$ higher than in ODF mode indicates that the RMS definition may different
for the two modes.

– Given that the RMS error determined by the TROPOMI processor is clearly different from the QDOAS results shows that
the QDOAS IF method calculates the RMS error differently from the TROPOMI IF method of Eq. (4).

As reference, Table 2 also includes the results of the regular TROPOMI retrieval of the currently officially available processor
version v1.2.2, as well as the results from a local reprocessing with the forthcoming v2.0.0 processor (Sect. 4.1.3). That
processor has an experimental option to also include an intensity offset correction, implemented in the form of an extra term
on the right hand side of Eq. (2):

$$R_{\mathrm{mod}}(\lambda) = P(\lambda) \cdot \exp[\dots] \cdot (\dots) + \frac{P_{\mathrm{off}}(\lambda) \cdot S_{\mathrm{off}}}{E_0(\lambda)} \ . \tag{10}$$

with $P_{\mathrm{off}}(\lambda)$ a low-order polynomial and $S_{\mathrm{off}}$ a suitable scaling factor with the same unit as $E_0(\lambda)$. Table 2 shows that including
a constant $P_{\mathrm{off}}$ in the TROPOMI retrieval has a similar effect as in the case of QDOAS: the GCD and the SCD error decrease
by a few percent.

Another small difference in the retrieval methods is that the TROPOMI NO$_2$ processor uses the level-1b uncertainty in $\chi^2$
minimisation (cf. Eq. (3)) whereas OMI/QA4ECV does not (cf. Eq. (7)). QDOAS has the option to turn the $\chi^2$ weighting on
in its ODF mode, the impact of which on the fit results (not shown) is minimal for the GCD and RMS, while the SCD error
seems to be unrealistically much reduced, indicating that perhaps the error propagation in the ODF mode is not done entirely
correctly.

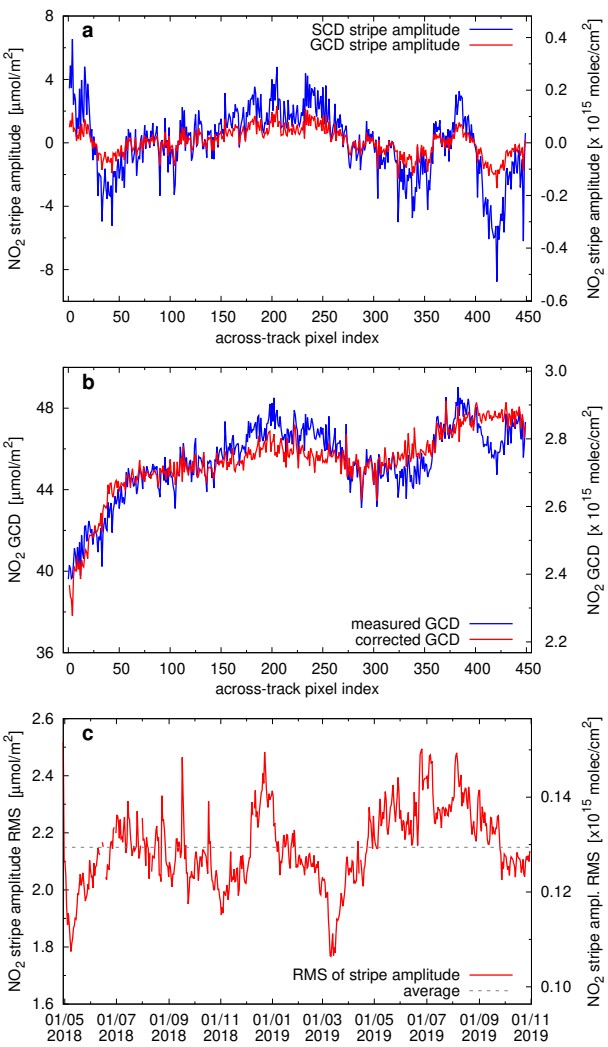

**Figure 4.** Evaluation of the NO$_2$ SCD stripe amplitude. **a)** SCD stripe amplitude $N_s^{str}$ (blue) and $N_s^{str}/M_{geo}$, i.e. the GCD stripe amplitude (red), for orbit 03711 of 1 July 2018. **b)** The measured (blue) and corrected (red) GCD for the same orbit, averaged over the TL range. **c)** Time evolution of the RMS of the SCD stripe amplitude.

All in all, the retrieval method itself (IF or ODF) does not seem to have a significant impact, while the intensity offset correction has quite a large impact on the GCD (and thus on the SCD) values. The intensity offset term is further discussed in Sect. 5.1.





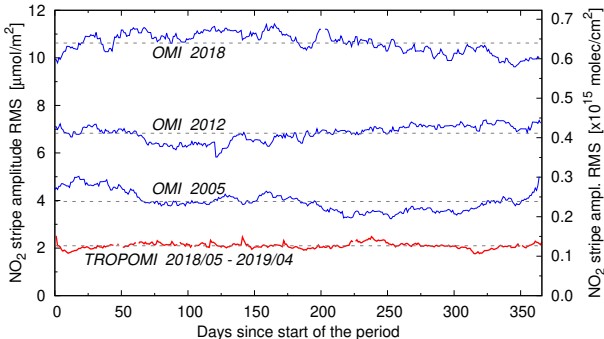

**Figure 5.** Comparison of the time evolution of the RMS of the $NO_2$ SCD stripe amplitude over the first year of TROPOMI data (red; cf. Fig. 4c) and over selected OMI/QA4ECV years (blue); the main increases in the OMI RMS occur during 2006, 2010-11 and 2014-15. Dashed lines indicate averages over the year periods.

### 4.3 De-striping: correcting across-track features

Since the beginning of the OMI mission, non-physical across-track variations in the $NO_2$ SCDs have been observed, which shows up as small row-to-row jumps or "stripes" (Boersma et al., 2011; Veihelmann and Kleipool, 2006). Given that the geophysical variation in $NO_2$ in the across-track direction (east-west) is smooth rather than stripe-like (Boersma et al., 2007), a

procedure to "de-stripe" the SCDs is implemented in the CTM/DA processing system used for DOMINO and QA4ECV. Even though in TROPOMI the row-to-row variation is much smaller than in OMI (cf. Fig. 3a), as of v1.2.0 it was decided to turn on de-striping to remove small but systematic across-track features and improve the data product quality.

The operational TROPOMI de-striping is determined from the TL range of orbits over the Pacific Ocean and a slant column stripe amplitude is determined for each viewing angle. The SCD stripe amplitude ($N_s^{str}$) is defined as the difference between

the measured total SCD ($N_s$) and the total SCD ($N_s^{corr} = N_s - N_s^{str}$) derived from the CTM/DA profiles using the averaging kernel and air-mass factor from the retrieval. In order to retain only features which are slowly varying over time, and in order to reduce the sensitivity to features observed during a single overpass, the SCD stripe amplitudes are averaged over a time period of 7 days, or about 7 Pacific orbits, before subtracting them from the SCDs. The $NO_2$ data product file contains $N_s$ and $N_s^{str}$, so that a user of the slant column data can/must apply the stripe correction.

As an example, Fig. 4a shows $N_s^{str}$ for the Pacific Ocean orbit of 1 July 2018 (blue) and $N_s^{str}/M_{geo}$ (red), the stripe amplitude in GCD space. For the same orbit Fig. 4b shows the GCD (blue) averaged over the TL range and the corrected GCD, i.e. $N_s^{corr}/M_{geo}$ (red). The across-track structure and the magnitude of the $N_s^{str}$ vary in time, but the overall behaviour is fairly constant.

A measure of the stability of the SCD stripe amplitude is the RMS of the across-track stripe amplitude. Fig. 4c shows this

RMS as function of time: there is quite some variation, but on average the RMS seems constant; further monitoring will have to show whether the stripe amplitude remains stable. Fig. 5 shows the same quantity for the first year of TROPOMI data (average: 2.14 $\mu$mol/m$^2$ = $0.13 \times 10^{15}$ molec/cm$^2$) and for selected years of OMI/QA4ECV data: 2005 (3.96 $\mu$mol/m$^2$ or 1.9 times





the TROPOMI average), 2012 (6.83 $\mu$mol/m$^2$ or 3.3 times), and 2018 (10.63 $\mu$mol/m$^2$ or 5.1 times). The increase in stripe amplitude of OMI NO$_2$ data is not uniform over time and is also present in case daily solar irradiance spectra are used for the retrieval (S. Marchenko, pers. comm., 2019), hence the increase is not (or at least not solely) caused by the use of a fixed irradiance in OMI/QA4ECV data (viz. Table 1),

## 5  4.4  Quantitative TROPOMI-OMI GCD comparison

The comparison of TROPOMI and OMI/QA4ECV Pacific Ocean orbits of 1 July 2018 in Fig. 3a is merely qualitative, because (a) of the row anomaly in the OMI data, (b) of the stripiness of the OMI data, and (c) the orbits do not exactly overlap. For a more quantitative comparison, TROPOMI and OMI data are gridded to a common longitude-latitude grid of $0.8° \times 0.4°$ – after applying the de-striping of the SCDs described in the previous subsection – and selecting (almost) cloud-free pixels only
(cloud radiance fraction $< 0.5$).

Fig. 6 shows the scatter plot of the TROPOMI and OMI/Q4ERCV GCDs for July 2018 for two regions: the remote Pacific Ocean and the polluted area covering India and China on the Northern Hemisphere. Both show a very good correlation with $R^2 \approx 0.99$. Over the Pacific Ocean area the TROPOMI GCD is on average $2.26 \pm 1.69$ $\mu$mol/m$^2$ ($1.35 \pm 1.01 \times 10^{14}$ molec/cm$^2$) or $5.23 \pm 3.93\%$ larger than OMI/QA4ECV. For Jan. 2019 (not shown) the TROPOMI GCD over the Pacific
is $2.19 \pm 1.56$ $\mu$mol/m$^2$ or $5.78 \pm 4.61\%$ larger than OMI/QA4ECV. Over the polluted India-to-China area the TROPOMI GCD is on average $2.02 \pm 2.08$ $\mu$mol/m$^2$ or $3.79 \pm 4.06\%$ larger than OMI/QA4ECV.

These differences between the TROPOMI and the OMI/QA4ECV GCDs (and thus between the SCDs) is comparable to the difference found in Sect. 4.2 due to turning on the intensity offset correction and may therefore be related mainly to the specific settings of the retrieval methods.

## 20  4.5  Impact of time difference between radiance and irradiance measurements

In the off-line TROPOMI NO$_2$ (re-)processing of a certain radiance orbit, the processor is configured to use the irradiance spectrum measured nearest in time to the radiance orbit. Given that TROPOMI takes irradiance measurements once every 15 orbits (once every $\sim$25h:22m) and that currently the off-line processing is running at least a week after the radiance measurements, the difference in time between the radiance and irradiance measurements will usually be not larger than 8 orbits. In this
sense, the TROPOMI processing is very different from the OMI processing (whether QA4ECV, OMNO2A or other): for OMI the 2005 average irradiance is used for the full dataset (2004-present) (van Geffen et al., 2015; Zara et al., 2018).

If for the TROPOMI processor one was to use a fixed irradiance, the errors on the retrieval results become larger. Fig. 7a illustrates this by showing the across-track TL range average SCD error for radiance orbit 07513 using irradiance measurement of the same orbit and of orbit 05428 (2085 orbits, 147 days earlier) and of orbit 03058 (4455 orbits, 314 days earlier): the larger
the difference in measurement time between radiance and irradiance, the larger the SCD error and the larger the row-to-row variation in the SCD error.

Fig. 7b shows the SCD error averaged over detector rows 25-424 (so as to avoid including the higher uncertainties of the outer rows related to the lower on-board pixel binning change) and corresponding standard deviation (stddev) for two radiance





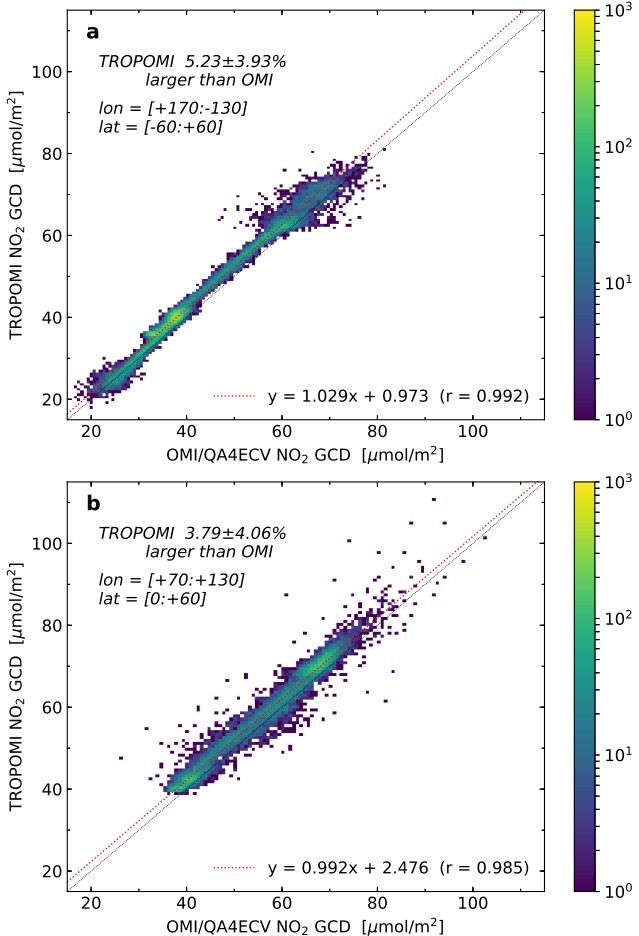

**Figure 6.** Comparison of TROPOMI and OMI/QA4ECV $NO_2$ GCD for July 2018 after conversion to a common longitude-latitude grid of $0.8° \times 0.4°$ for **a)** the Pacific Ocean and **b)** the India-to-China area. The area covered, the difference between TROPOMI and OMI/QA4ECV, the linear fit coefficients, and the correlation coefficient are listed in the panels.

orbits using selected irradiance measurements from between these two; in the case of radiance orbit 03058 (07513) future (past) irradiances are used. The average SCD error itself increases gradually with increasing time difference, while the stddev – a measure for the stripiness of the SCD error – increases more than linearly with time.

For the same series Fig. 7c shows that the average GCD value itself is not affected by the time difference between radiance and irradiance: for radiance orbit 03058 (07513) the average GCD is $41.11 \pm 0.18$ $\mu$mol/m$^2$ ($32.79 \pm 0.18$ $\mu$mol/m$^2$). The stddev of this averaging – the stripiness of the GCD – increases steeply, leveling off to a factor of around 3. If the TROPOMI processing were to use a fixed irradiance, the de-striping (Sect. 4.3) would show an ever increasing stripe amplitude in Fig. 4c.

It is unclear why the time difference between radiance and irradiance measurements has such a big impact on the TROPOMI $NO_2$ retrieval errors. The solar output varies somewhat over time, but it seems unlikely that this variation is large enough



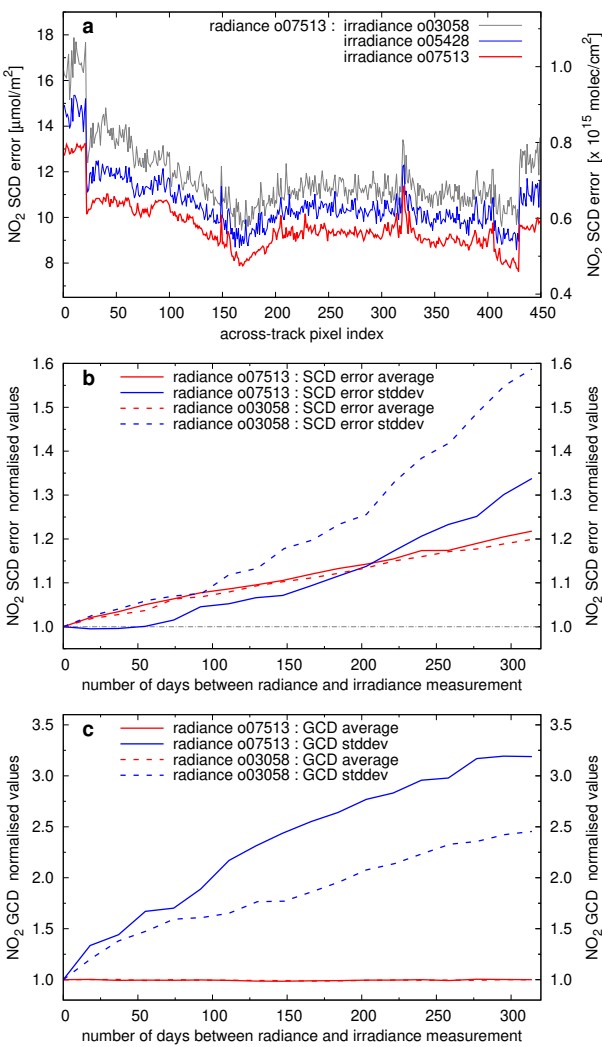

**Figure 7.** Effect of a difference between the radiance and irradiance orbit numbers on the NO$_2$ GCD and the SCD error, averaged over the TL range. **a)** SCD error of radiance orbit 07513 (26 March 2019; red) using irradiance measurements from orbits 03058 (16 May 2018; blue), 05428 (30 Oct. 2018; gray) and 07513. **b)** SCD error averaged over detector rows 25-424 (red) and corresponding standard deviation (blue) of two radiance orbits using a series of irradiance measurements, normalised to 1 for matching orbits, as function of the number of days between radiance and irradiance measurement. **c)** Idem for the GCD (red) and corresponding standard deviation (blue).

to cause the increase in the retrieval errors. TROPOMI suffers from a small degradation (Rozemeijer and Kleipool, 2019) of $1-2\%$, notably in the irradiance, but with little to no wavelength dependency, hence this degradation is not expected to significantly affect the reflectance and the NO$_2$ SCD retrieval results.

The increased stripiness observed in the OMI NO$_2$ results depicted in Fig. 5, and shown by Boersma et al. (2011) and
5  discussed in detail by Zara et al. (2018), is at least in part the result of the increasing difference in time between radiance and

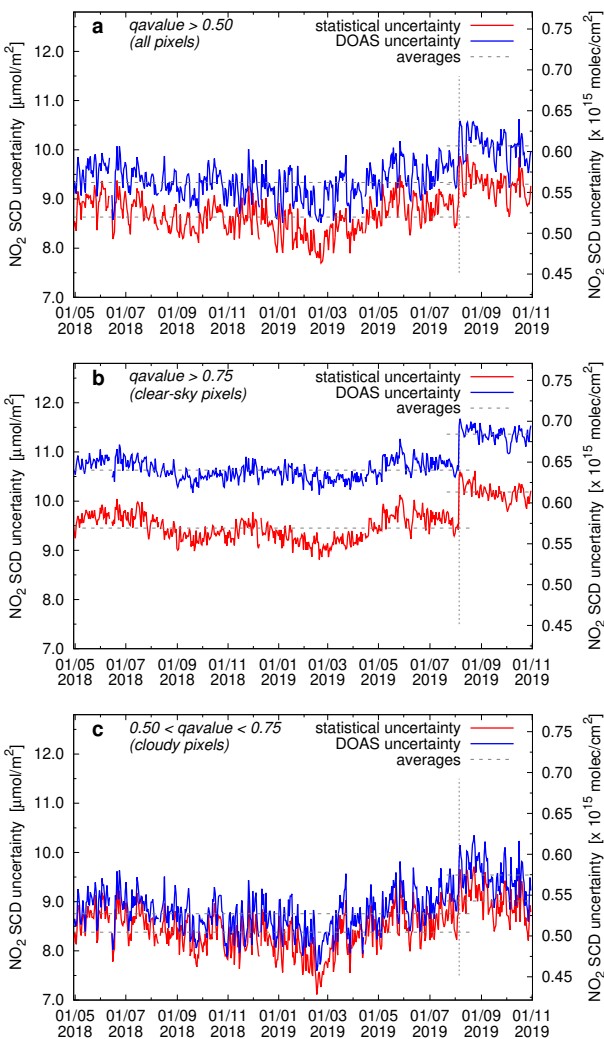

**Figure 8.** NO₂ SCD statistical uncertainties (red) and SCD error estimates from the DOAS fit (blue) as function of time. **a)** All pixels with successful retrieval. **b)** Pixels with cloud radiance fraction < 0.5. **c)** Pixels with cloud radiance fraction > 0.5. The vertical dotted line marks 6 Aug. 2019, when the along-track ground pixel size was reduced. Averages, marked by dashed lines, are listed in Table 3.

irradiance meeasurement, but acting over a longer time scale than the effect seen in Fig. 7b-c for TROPOMI. The fact that the GCD value itself (Fig. 7c) is not appreciably affected by the time difference is very reassuring, both for the TROPOMI and the OMI/QA4ECV retrieval results.





### 4.6  Time dependence of the slant column uncertainty

The spatial variability of the SCDs over a remote Pacific Ocean sector can be used as an independent statistical estimate of the random component of the SCD uncertainty. This approach was used in the QA4ECV project by Zara et al. (2018) to compare OMI and GOME-2A $NO_2$ and formaldehyde SCD values retrieved by different retrieval groups, as well as to compare the SCD
error estimates following from the different DOAS fits.

Fig. 8 shows the $NO_2$ SCD statistical uncertainties (red) and SCD error estimates from the DOAS fit (blue) as function of time for all ground pixels for which the retrieval was succesfull (i.e. with quality assurance value `qa_value` $> 0.50$; top panel), for clear-sky pixels (`qa_value` $> 0.75$, corresponding to cloud radiance fraction $< 0.5$; middle panel), and for cloudy pixels ($0.50 <$ `qa_value` $< 0.75$; bottom panel). For this exercise the Pacific Ocean orbits (Sect. 2.1.2) where evaluated
over the latitude range $[-60° : +60°]$. Averages over the data period shown in Fig. 8 are listed in Table 3, along with the OMI/QA4ECV results from Zara et al. (2018), who also showed that the OMI/QA4ECV SCD statistical uncertainties and SCD error estimates have increased over the years by about $1\%$ and $2\%$ per year, respectively.

The reduction of the along-track ground-pixels size from $7.2$ km to $5.6$ km on 6 Aug. 2019 effectively entails a reduction of the integration time from $1.08$ s to $0.84$ s, as a result of which the per-pixel noise on the level-1b radiances data increased by a
factor of $\sqrt{1.08/0.84} = 1.134$, which in turn caused an increase of the $NO_2$ SCD error by somewhat less than $13\%$ (because the SCD error is not solely determined by the noise on the radiance spectra). This increase in the SCD error is visible in Fig. 8 as a jump at that date (marked by a vertical dotted line), and is reflected in the averages given in Table 3: the DOAS uncertainty increases by $7 - 10\%$, depending on the pixel type. The pixel size change does not impact the average magnitude of the $NO_2$ GCD (except for polluted regions where due to the smaller pixels size larger peak values may be expected), but it does have
an effect on the inter-pixel variation of the GCD: the statistical uncertainty increases by $\sim 9\%$, almost independent on the pixel type.

All in all, the TROPOMI statistical uncertainties are clearly much lower than those of OMI/QA4ECV, even after the ground pixel size reduction. The SCD error estimates from the DOAS fit routine are on average larger than the statistical uncertainties (for TROPOMI about $10\%$ and for OMI/QA4ECV about $20\%$). From the TROPOMI data it appears that the SCD uncertainty
is only $4\%$ larger than the statistical uncertainty in case of cloudy pixels, but about $12\%$ in case of clear-sky pixels. The main reason for the difference between the DOAS and statistical uncertainties is that, unlike the statistical uncertainties, the SCD error estimates also include systematic retrieval issues, and these appear to play a larger role for clear-sky pixels, i.e. pixels for which the radiance signal is lowest. From Fig. 8 it is furthermore clear that the statistical and the DOAS uncertainties of TROPOMI appear to be stable over the currently available data period.





**Table 3.** NO$_2$ SCD statistical and SCD DOAS fit uncertainties, averaged over the listed period, given in two units; cf. Fig. 8.

|  | TROPOMI 2018/04/30 2019/08/05 | TROPOMI 2019/08/06 2019/10/31 | OMI [a] 2005/01/01 2015/12/31 |
|---|---|---|---|
| *unit = $\mu$mol/m$^2$* | | | |
| *all pixels* | | | |
| statistical | $8.63 \pm 0.34$ | $9.30 \pm 0.28$ | 11.45 |
| DOAS | $9.33 \pm 0.33$ | $10.07 \pm 0.30$ | 13.87 |
| *clear-sky pixels* | | | |
| statistical | $9.45 \pm 0.25$ | $10.19 \pm 0.18$ | 12.64 |
| DOAS | $10.63 \pm 0.19$ | $11.36 \pm 0.15$ | 15.11 |
| *cloudy pixels* | | | |
| statistical | $8.38 \pm 0.40$ | $9.01 \pm 0.35$ | 10.88 |
| DOAS | $8.75 \pm 0.40$ | $9.54 \pm 0.38$ | 13.91 |
| *unit = $10^{14}$ molec/cm$^2$* | | | |
| *all pixels* | | | |
| statistical | $5.20 \pm 0.20$ | $5.60 \pm 0.17$ | 6.89 |
| DOAS | $5.62 \pm 0.20$ | $6.07 \pm 0.18$ | 8.36 |
| *clear-sky pixels* | | | |
| statistical | $5.69 \pm 0.15$ | $6.13 \pm 0.11$ | 7.61 |
| DOAS | $6.40 \pm 0.11$ | $6.84 \pm 0.09$ | 9.10 |
| *cloudy pixels* | | | |
| statistical | $5.05 \pm 0.24$ | $5.43 \pm 0.21$ | 6.55 |
| DOAS | $5.27 \pm 0.24$ | $5.75 \pm 0.23$ | 8.38 |

[a]) OMI/QA4ECV results taken from Zara et al. (2018), Table 4; additional
data provided by the author.

## 5 Discussion

### 5.1 Intensity offset correction

Many DOAS applications, including the OMI/QA4ECV processing, include a correction for an intensity offset in the radiance,
e.g. in the form given in Eq. (9). The precise physical origin of such an intensity offset is not specified in the literature, but it
5 is thought to be related to instrumental issues (e.g. incomplete removal of straylight or dark current in level-1b spectra) and/or
atmospheric issues (e.g. incomplete removal of Ring spectrum structures, vibrational Raman scattering in clear ocean waters);
see, for example, Platt and Stutz (2008), Richter et al. (2011), Peters et al. (2014), Lampel et al. (2015).





From OMI/QA4ECV evaluations (Müller et al., 2016; Boersma et al., 2018) and a preliminary study using TROPOMI data (Oldeman, 2018) it appears that the largest impact of the intensity offset correction occurs over cloud-free clear ocean water (i.e. with little to no chlorophyll). If indeed absorption by VRS is the key aspect here, it would on physical grounds be more appropriate to include a VRS absorption spectrum ($\sigma_{VRS}$) in the DOAS fit, because the intensity offset corrections is proportional to the irradiance, while $\sigma_{VRS}$ has a different spectral structure, i.e. an intensity offset correction will not fully compensate for VRS absorption. Investigating this matter further falls outside the scope of the present paper.

Turning on the intensity offset correction ("IOC") in QDOAS for the TROPOMI and OMI orbits shown in Fig. 3 reduces the GCD values on average by $\sim 5\%$, with the relative impact largest for the lower GCDs. Since this decrease of the GCDs is comparable for both TROPOMI and OMI data, when using the same SCD processor, it seems unlikely that the IOC is correcting for instrumental effects. It must be noted that the effect of the IOC in QDOAS (viz. Eq. (9)) on the GCDs is nearly twice as large as the effect of the experimental IOC in the TROPOMI processor (viz. Eq. (10)); apparently these two implementations of the IOC do not behave exactly the same.

All in all an intensity offset correction will not be included in the regular TROPOMI $NO_2$ processing, also because instrumental effects such as straylight and dark current are adaquately corrected for in the spectral calibration in the level 0-to-1b processor.

## 5.2 Validation of stratospheric $NO_2$

Routine validation of TROPOMI data products is being carried out by the Validation Data Analysis Facility (VDAF; http://mpc-vdaf.tropomi.eu/), with support from the S5P Validation Team (S5PVT), which issues Quarterly Validation Reports, such as Lambert et al. (2019). Since $NO_2$ over the Pacific Ocean is primarily stratospheric $NO_2$, validation of stratospheric $NO_2$ essentially is also validation of Pacific Ocean $NO_2$ SCDs.

Stratospheric $NO_2$ column data are compared to reference measurements from Zenith-Sky Light (ZSL) DOAS instruments, which are operated in the context of the Network for the Detection of Atmospheric Composition Change (NDACC). ZSL-DOAS measurements, obtained twice daily at twilight, are adjusted to the TROPOMI overpass time in order to account for the diurnal cycle of $NO_2$. Quoting the 4th Quarterly Validation Report (Lambert et al., 2019), the TROPOMI stratospheric $NO_2$ columns are "generally lower by approximately $0.25 \times 10^{15}$ molec/cm$^2$ [4 $\mu$mol/m$^2$] than the NDACC ZLS-DOAS ground-based measurements, deployed at 14 stations from pole to pole. The bias of roughly $11\%$ is thus slightly above the S5P mission requirements of $10\%$, which is equivalent to $0.2 - 0.4 \times 10^{15}$ molec/cm$^2$, depending on latitude and season." The $-11\%$ bias mentioned is the average bias; the median bias is about $-7\%$. Note that the ZSL-DOAS measurements have their own uncertainties (a bias of at most $10\%$ and a random uncertainty better than $1\%$; Lambert et al., 2019), and that the interpolation to the TROPOMI overpass time introduces uncertainties in the ground-based data of the order of $10\%$ (Lambert et al. (2019); see also Dirksen et al. (2011)).

In other words: the agreement between stratospheric $NO_2$ of TROPOMI and ground-based instruments is rather good, where TROPOMI seems to give SCD column values that are slightly too low. Including an intensity offset correction in the DOAS fit





(Sect. 5.1) would lead to a reduction of the Pacific Ocean $NO_2$ SCD by a few percent (Sect. 4.2), which in turn would imply worsening of the validation results.

### 5.3  $NO_2$ retrieval over strongly polluted areas

In case $NO_2$ concentrations are no longer optically thin, assumptions lying at the basis of the DOAS retrieval approach may no
longer be valid (Richter et al. (2014); A. Richter, pers. comm., 2019): the relationship between SCD and VCD may become non-linear for single wavelengths, the AMF of boundary layer $NO_2$ may become strongly wavelength dependent and decrease with increasing $NO_2$ columns, and the temperature dependence of the $NO_2$ reference spectrum (usually corrected for a-posteriori in the AMF application) may show spectral structures. During a dramatic pollution episode in China in January 2013, with $NO_2$ up to $1 \times 10^{17}$ molec/cm$^2$ (1660 $\mu$mol/m$^2$), these effects seemed to become significant, as shown by Richter et al. (2014).
When measuring $NO_2$ over strongly polluted areas with high spatial resolution, such as provided by TROPOMI, the chance of detecting very large $NO_2$ concentrations for individual ground pixels increases. The area with largest $NO_2$ columns is probably China, but since the reductions in air pollution in China over the past years, it is currently unlikely to encounter $NO_2$ concentrations that are not optically thin in the TROPOMI measurements. In July 2018, for example, the largest number of ground pixels with a GCD exceeding 300 $\mu$mol/m$^2$ ($2 \times 10^{16}$ molec/cm$^2$) is 453 (0.04%) of the 1227234 pixels with a
successful retrieval in orbit 03846 (11 July) over Africa, notably occuring in two patches over the Highveld region in South Africa (141 pixels, 10 of which are cloudy) and along the border between Angola and DR Congo (310 pixels, 6 cloudy). The highest GCD found for this orbit is $883 \pm 16$ $\mu$mol/m$^2$; there are 6 ground pixels with a GCD exceeding 600 $\mu$mol/m$^2$.

### 6  Concluding remarks

This paper documents the $NO_2$ slant column density (SCD) retrieval method in use for TROPOMI measurements and discusses
the stability and uncertainties of the retrieval results. The SCD is key input to the next steps in the $NO_2$ processing chain: the determination of the tropospheric and stratospheric $NO_2$ vertical column densities. Knowledge of the quality and the stability of the SCD retrieval results is therefore important by itself.

The TROPOMI $NO_2$ SCD retrieval describes the modelled reflectance in terms of a non-linear function of the relevant reference spectra and uses Optimal Estimation to minimise the difference between the measured and modelled reflectance. The
results of this retrieval method compare very well with SCD retrievals performed with the QDOAS software (Danckaert et al., 2017) when using settings as close as possible to those of the TROPOMI processor.

The SCD statistical uncertainty originating from the local variability of the SCD over the Pacific Ocean (a remote, source-free region) and the uncertainty estimate following from the DOAS retrieval are quite stable over time. The TROPOMI statistical uncertainties are lower by about 30% (20% since the ground pixel size reduction on 6 Aug. 2019) than those of OMI/QA4ECV
(Zara et al., 2018), and the SCD error estimates from the DOAS fit routine are on average larger than the statistical uncertainties: for TROPOMI about 10%, but for OMI/QA4ECV about 20%. The along-track pixel size reduction from 7.2 km to 5.6 km on 6 Aug. 2019 has resulted in an increase of the DOAS and statistical uncertainties by about 10%.





Quantitative comparison against OMI/QA4ECV data (i.e., OMI measurements processed within the QA4ECV project; Boersma et al., 2018) over the full Pacific Ocean shows very good agreement with a correlation coefficient better than $0.99$. TROPOMI values are, however, about $5$ $\mu$mol/m$^2$ or $5\%$ higher than the OMI/QA4ECV values, which seems to be due mainly to the fact that the OMI/QA4ECV processing includes a so-called intensity offset correction, which is not applied in the

TROPOMI processing: retrieval of TROPOMI data using QDOAS with different settings shows that the intensity offset correction reduces the SCDs by $4.5 - 5.0\%$.

Since NO$_2$ over the Pacific Ocean is primarily stratospheric NO$_2$, validation of stratospheric NO$_2$ essentially is also validation of Pacific Ocean NO$_2$ SCDs. As reported by Lambert et al. (2019), TROPOMI stratospheric columns are lower than ground-based measurements by about $4$ $\mu$mol/m$^2$ ($0.25 \times 10^{15}$ molec/cm$^2$). Since the introduction of an intensity offset cor-

rection reduces the SCD by a few percent, it would thus worsen the validation result. Because the physical nature of such an intencity offest is unclear, there are no plans to include an intensity offset correction in future updates of the TROPOMI NO$_2$ SCD retrieval.

The non-physical row-to-row variation (stripe amplitude) of the TROPOMI SCDs (on average $2.14$ $\mu$mol/m$^2$) is much lower than in the case of OMI/QA4ECV (in 2005 $\sim 2$ and in 2018 $\sim 5$ times the TROPOMI average) but even so a so-called destriping

of the TROPOMI SCDs is applied.

In view of both the SCD error estimate and the across-track striping of the SCDs, it is essential to use an irradiance spectrum measured as close as possible in time to the radiance measurement in the DOAS fit: the larger the time difference between these two, the larger the SCD error and the larger the stripiness.

## Appendix A:  Implementation of the Ring correction
**in the intensity and optical density fit models**

An essential difference between the IF retrieval for TROPOMI and the retrieval with QDOAS, whether in ODF mode or IF mode, is the implementation of the correction for the Ring effect, where the authors believe that the TROPOMI implementation is physically more accurate.

In the case of the TROPOMI retrieval (and OMI retrieval using OMNO2A) the correction is included as a non-linear term

in the modelled reflectance – the term between large parenthesis in Eq. (2) – which depends on a modelled Ring reference spectrum ($I_{\text{ring}}$) and the measured irradiance ($E_0$).

In the case of QDOAS (and similar retrieval algorithms of other institutes) the correction is included as a linear term in the form of a pseudo-absorber in the modelled reflectance – the last term in Eq. (6) – which depends on a fixed reference spectrum determined from a modelled Ring reference spectrum and a convolved reference irradiance spectum ($\sigma_{\text{ring}} = I_{\text{ring}}/E_{\text{ref}}$ minus

a 2nd order polynomial).

The terms on the right hand side in Eq. (2) can be written as $\exp(Y) \cdot (1+x)$. Taking the natural logarithm and using a Taylor expansion gives $\ln[\exp(Y) \cdot (1+x)] = Y + \ln(1+x) = Y + x - x^2/2 + x^3/3 - \ldots$ In other words, Eq. (2) reduces to Eq. (6)



in case $x \ll 1$, which is usually the case since $|C_{\mathrm{ring}}|$ is less than $0.075$ for most ground pixels, assuming $I_{\mathrm{ring}}/E_0$ and $\sigma_{\mathrm{ring}}$ are the same.

In terms of the cases listed in Table 2, the retrieval of QDOAS case 6 is closest to the TROPOMI retrieval (case b). For all pixels with valid retrieval $C_{\mathrm{ring}}^{(6)} = 0.924 \cdot C_{\mathrm{ring}}^{(b)} + 0.001$, with a correlation coefficient better than $0.999$. Absolute differences between the coefficients range from $-0.002$ to $+0.006$, with largest differences over ocean areas without clouds; above clouds the differences are a factor of $10$ smaller. These differences are probably related to the use of the measured or the modelled irradiance spectrum, but the effect on the fit results seems to be quite small. ($C_{\mathrm{ring}}$ results from QDOAS case 1 differ slightly from case 3, with a difference smaller than the differene between case 1 and case b.)

## Appendix B:  Relationship between the RMS error

## in the intensity and optical density fit model

The RMS error of the intensity fit, given in Eq. (4), and of the optical density fit, Eq. (8), are defined differently, but a first order relationship between the two can be derived as follows (A. Richter, pers. comm., 2019).

For good fits the ratio $R_{\mathrm{meas}}/R_{\mathrm{mod}} \approx 1$ and since $\ln(x) - \ln(y) = \ln(x/y) \approx x/y - 1$ for $x/y \approx 1$, the summation in Eq. (8) can be re-written as: $\sum((R_{\mathrm{meas}} - R_{\mathrm{mod}})/R_{\mathrm{mod}})^2$. For not too strongly varying modelled reflectances this can be approximated by: $1/\overline{R_{\mathrm{mod}}^2} \cdot \sum(R_{\mathrm{meas}} - R_{\mathrm{mod}})^2$. With this, the ratio between the RMS values of the two methods is: $R_{\mathrm{RMS}}/R_{\mathrm{RMS}}^{\mathrm{ODF}} \approx \left(\overline{R_{\mathrm{mod}}^2}\right)^{1/2} \approx \overline{R_{\mathrm{mod}}}$, since the root-mean-square of the modelled reflectance can be approximated by the average modelled reflectance.

For the ground pixels with a good quality fit (`qa_value` $\geq 0.5$) of an arbitrary TROPOMI orbit the ratio between the RMS values appears to agree with the average modelled reflectance to within $3.7\%$.

## Appendix C:  TROPOMI spike removal

In order to remove strong outliers in the DOAS fit residual (caused by, e.g., high-energy particles hitting the CCD detector, variations in the dark current, or pixels not correctly flagged in the level-1b data in case of over-exposure due to clouds), a "spike removal" algorithm will be used as of v2.0.0 (cf. Sect. 4.1.3). After removal of such outliers from the measured reflectance, the DOAS fit is redone to provide the final fit parameters, which is not followed by another check on outliers, to avoid ending up in a cycle. Outliers occur only in a small fraction of the ground pixels: usually $\sim 5\%$ of the successfully processed ground pixels, most of which have less than $5$ outliers per ground pixel; the largest effects will occur over the South Atlantic Anomaly (where the impact of high-energy particles on the detector occurs frequently; cf. Richter et al. (2011)) and over bright clouds (where saturation occurs frequently). Hence, the results presented in this paper are not expected to change significantly by the introduction of the spike removal.





The algorithm implemented in the $NO_2$ SCD retrieval for the removal of outliers in the fit residual (van Geffen et al., 2019, App. F) uses the box-plot method[2], which determines lower and upper values based on the first and third quartiles, $Q_1$ and $Q_3$, i.e. the 25th and 75th percentile of a distribution (the second quartile, $Q_2$, is the median). If a certain value is larger than $Q_3 + Q_f \cdot Q_{3-1}$ or lower than $Q_1 - Q_f \cdot Q_{3-1}$, with $Q_{3-1} = Q_3 - Q_1$ the inter-quartile range and $Q_f$ a suitable

multiplication factor, it is termed an outlier. The so-called inner and outer fences have $Q_f = 1.5$ and $Q_f = 3.0$, respectively. For the TROPOMI $NO_2$ SCD v2.0.0 retrieval the outer fences will be used as criterion for outlier detection.

*Author contributions.* JvG conducted the research described in this paper and is responsible for the text. MS and MtL implemented and tested the retrieval code in the TROPOMI processor. HE and KFB are responsible for the final $NO_2$ data product. MZ has been involved in the uncertainty estimates. PV has been involved in retrieval issues and is the PI of TROPOMI.

*Competing interests.* There are no competing interests.

*Acknowledgements.* The authors would like to thank the following people for discussions on retrieval issues: Piet Stammes, Johan de Haan, Andreas Richter, Steffen Beirle, Michel Van Roozendael, Sergey Marchenko. And the following people for discussions on level-1b issues: Quintus Kleipool, Nico Rozemeijer, Antje Ludewig and Erwin Loots. Sentinel-5 Precursor is a European Space Agency (ESA) mission on behalf of the European Commission (EC). The TROPOMI payload is a joint development by ESA and the Netherlands Space Office (NSO).

The Sentinel-5 Precursor ground-segment development has been funded by ESA and with national contributions from The Netherlands, Germany, and Belgium. Contains modified Copernicus Sentinel data 2018-2019.

---

[2] "What are outliers in the data?"; https://www.itl.nist.gov/ div898/handbook/prc/section1/prc16.htm



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
