# Peer review of "S5P/TROPOMI NO$_2$ slant column retrieval: method, stability, uncertainties, and comparisons against OMI"

_Atmospheric Measurement Techniques, 2019_

## Referee Comment (RC1) · Anonymous Referee #1 · 13 Jan 2020

Any early assessment of the data quality for a major remote-sensing mission (such as TROPOMI) is always welcome, and the study in review, "S5P/TROPOMI NO 2 slant column retrieval: method, stability, uncertainties, and comparisons against OMI", by van Geffen et al., delivers a timely and conscientious report on the particulars of the NO2 slant column density retrievals from the TROPOMI data. The article is well-written and reasonably compact. I perceive no major problems that may delay a prompt publication. My comments are mostly of a clarifying or editorial nature.

Specific comments:

P.8,l.11: The smaller-pixel size doubles the percentage of flagged NO2 retrievals. This

leads to additional ~2*10ˆ3 flags/orbit compared to the larger-pixel case. Please comment, if you may, on the possible cause of this doubling. Are these flagged retrievals randomly distributed across the orbit? Are they mostly related to the saturated pixels?

P.9,l.16. Please comment on the evolution of the radiance-irradiance wavelength difference along the TROPOMI orbit.

P.9,l.23: "A similar seasonal variation, though larger in magnitude, is seen in the OMI wavelength calibration data...". Fig. 34 from Schenkeveld et al. (2017) shows comparable (early-mission OMI data) magnitudes of the seasonal $<w_s>$ variability, without any detectable long-term trends in the OMI VIS channel. Please re-phrase ll. 23-27 accordingly.

P.11, Eq. (9) As written, the intesity offset implies a [potentially] non-linear change of the wavelength sampling for radiances. Please check.

P.12,l.1. "For OMI/QA4ECV both a shift and stretch are fitted;..." Is the stretch term essential for OMI? Please comment.

P.12, Sect.3.4 - Please specify whether OMNO2A v2.0 uses the intensity offset term.

P.12, Sect.4.1, 1st par. - Please comment on how representative this particular TROPOMI orbit is. Have similar comparisons been done for different seasons? Can the shown trends be safely extrapolated on a ~year-round sample? This in particular applies to the tests shown in Fig.3b,e. To some extent, this question is answered in P.19. It is worthwhile to make a general statement earlier on.

P.14, last par. Perhaps it would be worthwhile to point to the big difference in the striping patterns delivered by various OMI algorithms, also mentioning that the forthcoming Sect. 4.3 quotes exclusively the OMI/QA4ECV stats.

P.15,l.5. "The increased SCD error visible in the TROPOMI data of Fig. 3d-e around $\theta \approx +20$deg is related to the presence of saturation effects above bright clouds along this particular orbit." Nevertheless, most of the corresponding saturation-affected GCD values seem normal. Does this imply that the 405-465nm retrieval range retains enough of the saturation-free data to provide ~normlal-looking SCDs? Is there anything else helping to stabilize the saturation-affected retrievals? Please comment on.

P.16,l.18. The 4-times difference in the QDOAS/TROPOMI RMS estimates is hard to overlook, indeed. Considering the typical S/N~1500 in the TROPOMI data, one may side with the RMS~8*10ˆ(-4) provided by QDOAS. Similar- magnitude RMS~0.5-1.0*10ˆ(-4) is frequently quoted by various groups working with ~similar S/N data sets from various spacecrafts. The cited (Table 2) RMS~0.2*10ˆ(-4) seems like an overly optimistic assessment. Any comment?

P.18,l.19 "Fig. 4c shows this RMS...". Please provide more details on how this value was calculated.

P.19,l.9 Is de-striping applied both to TROPOMI and OMI? Please clarify in the text.

Fig.6. Since the intensity offset has the highest impact over the cloud-free, clear-water ocean areas (Sect. 5), one may test this by segregating the data in Fig.6a into two cases (cloudy and cloud-free; probably, selecting even more extreme cases than in Fig.8) and commenting on. I consider this as an important test, in light of the findings from Oldeman (2018).

P.22,l.2 "The fact that the GCD value itself (Fig. 7c) is not appreciably affected by the time difference is very reassuring..." Actually, I find this really puzzling: cf. Fig.7b and 7c. The SCD error, as anticipated, linearly increases in time in Fig.7b, but remains essentially flat in Fig.7c. Please help me (and the readers) to interpret this.

P.23,l.12 Table 3: Please specify whether the TROPOMI (presumably, yes) and OMI (?) SCDs were de-striped.

P.23,l.28 "From Fig. 8 it is furthermore clear that the statistical and the DOAS uncertainties of TROPOMI appear to be stable over the currently available data period." This is not what Fig.8 shows (the Aug. 6 jump). Please re-phrase.

P.25,l.14 "...also because instrumental effects such as straylight and dark current are adequately corrected for in the spectral calibration in the level 0-to-1bprocessor." Either remove the statement or provide the references that address the subject and are based on assessment of the post-launch data (both radiances and irradiances).

Section 5.2 I have a problem linking the statement "Since NO2 over the Pacific Ocean is primarily stratospheric NO2, validation of stratospheric NO2 essentially is also vali-dation of Pacific Ocean NO2 SCDs." Besides the point that it should be re-phrased, I do not see any connections between the discussed TROPOMI retrievals over Pacific and the ZSL-DOAS/SAOZ S5PVT network that completely avoids the Pacific basin. If the authors employed somewhat different approach than in Lambert et al. (2019), then they should provide a detailed description of the validation process. If this Section provides a summary of the Lambert et al. report and nothing besides, it should say so.

P.26,l.7 "...and the temperature dependence of the NO2 reference spectrum (usually corrected for a-posteriori in the AMF application) may show spectral structures..." Do you imply: "...and the temperature correction of the NO2 VCDs (usually introduced a-posteriori in the AMF application) may result in spectral artifacts in the fitting residuals that are linked to the temperature dependence of the NO2 reference spectrum." ? The NO2 reference spectrum always shows spectral structures in the 405-465nm range...

P.27,l.7 "Since NO2 over the Pacific Ocean is primarily stratospheric NO2, validation of stratospheric NO2essentially is also vali-dation of Pacific Ocean NO2SCDs. As reported by Lambert et al. (2019..." Again, I fail to relate the discussed Pacific Ocean retrievals to the Lambert et al. report.

P.28,l.3 There is no QDOAS Case 6 in Table 2. Please correct.

P.28,l.23 "After removal of such outliers..." Please specify how the removal is done: are the pixels corrected and re-used or completely removed from the fit?

P.28,ll.25 If this concerns the total, spatial x spectral number (I presume, though, that

the authors speak of the spatial domain only – please clarify!), $\sim$5% is, actually, a sizable population of the pixels. It may grow in time, eventually leading to much noisier and probably biased (depending on the preferential location of the spikes) retrievals. Any comments on this?

P.28,l.26 "...most of which have less than 5 outliers per ground pixel..." Please clarify that these 5 outliers happen in the spectral domain.

Technical corrections:

P.4,l.23: "...away from anthropogenic sources of NO2..."

P.9,l.28: "The dominant term in the overall magnitude of the radiance is the inhomogeneous illumination..." This needs to be clarified: "For a given field-of-view (row), the dominant term in the overall magnitude..." - is this what the authors meant?

P.9, l.29 "The magnitude of the day-to-day variation in the average is much smaller than this long-term oscillation..." By 'this long-term oscillation' do you mean ...the seasonal (Fig.2b) oscillation...?

P.15,l.19 "...the TROPOMI processor reports 10.2% ..."

P.15,l.27 "TROPOMI level-1b version 1.0.0 spectra..."

P.16,l.6 "...with other configuration settings as much as possible matching those of the TROPOMI processor..."

P.16,l.16 "...that the RMS definition may be different..."

P.16,l.20 "As a reference..."

P.18,l.4 "... is smooth rather than stripe-like over the non-contaminated areas..."

P.25,l.4 "...the intensity offset corrections are..."

P.25,l.14 adequately

---

## Referee Comment (RC2) · Anonymous Referee #2 · 23 Jan 2020

The manuscript "S5P/TROPOMI NO2 slant column retrieval: method, stability, uncertainties, and comparisons against OMI" provides a detailed description of the DOAS slant column fitting for TROPOMI NO2 measurements and discusses important issues related to the method, instrument, and verification. The stability and uncertainty analysis is particularly interesting. The manuscript is well organized and written. I recommend publication after a few minor revisions.

P4 L7 Is there a particular reason that the VIS band stops at 496 nm? While quite a few literatures numbered the TROPOMI VIS band until 500 nm.

P4 L20 Please give the full name of ATBD.

P4 S2.1.2 Please give a short description of the saturation (perhaps also blooming) effect for TROPOMI, which is mentioned and analyzed in the result section and important for TROPOMI NO2 measurements.

P5 L12 What would be the implication of increasing the size towards the edge when comparing the OMI and TROPOMI measurements?

P6 Table 1 I recommend to add the retrieval processor, namely TROPOMI (if no other name) vs QDOAS, before the date version. Please also add information of OMI/OMNO2A (almost same as TROPOMI) in the title or as table footnote etc.

P9 L23 Please give rough numbers of the magnitude for OMI?

P11 Eq 9 The intensity offset term shall not be placed in the same parentheses with lamda.

P12 L10 Please give the full name of VCD.

P12 L15 What is the implication and how large is the impact of changing the LM solver to OE method with Gauss-Newton?

P14 L17 & P15 L20 Following the previous question, is the difference between TROPOMI and OMNO2A (Fig 3c and 3f) mainly resulted from the difference in the wavelength calibration window (I assume not the mathematic solver)?

P15 L6 Previous introduction has mentioned that TROPOMI measurements suffering from saturation are filtered out. But the residual saturation is still affecting the retrieval (even not strongly) in Fig 3d-e. Is there any recommendation of further removing the effect during retrieval or data using?

P25 L3 Please give the full name of VRS (introduced already in Sect 3).

P26 L13 Even the air pollution is reduced in China, it is still very possible to see columns of a few 10e16 molec/cm2 (not optically thin), particularly in Winter. With this pollution level, the boundary AMF will also show spectral features, and this nonlinearity will
contribute a few percent difference to the retrieval during pollution episode. Please rewrite the statements and perhaps also provide the example for Winter for China (at the moment is July for Africa).

P28 L3 Do you mean case QDOAS case 4?

**AMTD**

---

## Referee Comment (RC3) · Anonymous Referee #3 · 24 Jan 2020

General Comments:

This paper describes the slant column density retrieval for the TROPOMI satellite instrument and gives a detailed assessment of uncertainties and comparisons with OMI. The paper is well written and despite the topic being very technical, I found the paper very clear and easy to follow. The details will be of interest to a limited set of scientists (retrieval algorithm developers mostly) but it is a thorough record of the uncertainties and preliminary temporal behaviour of the TROPOMI slant column retrievals. I recommend it be published following a few minor revisions.

The paper focussed almost entirely on an analysis of slant columns from tropical observations over the clean remote Pacific. I would have liked to have seen some slant column retrieval comparisons over an entire orbit, as some of the differences between retrieval approaches or instruments might be exaggerated at the highest solar zenith angles. Perhaps adding just a sentence or two describing how well the conclusions drawn about uncertainties, retrieval algorithm comparisons etc extend to cases other than the remote Pacific could be useful.

Specific Comments:

Page 3, Line 11: Give units of conversion factor.

Page 5, Line 31: I'm a bit confused by the wording describing a satellite latitude range. How is this changing between these two end points of 1 Jan and 1 July?

Page 9, Line 25: Comparing to OMI but no OMI results shown, so could you give a number indicating the magnitude of OMI variations?

Page 19, Line 12: define "India and China" latitude/longitude region

Page 21, Line 2: degradation of 1-2% relative to what? Is this degradation in throughput per year?

Figure 7: I find the colors of b and c very hard to follow in my mind. I think it's more common to be looking at a solid line that represents the average and a dotted line of the same color that represents a standard deviation or similar. Here they are different colors but the same pattern for a single orbit (backwards to what I'm used to). Not a significant issue but I just find it a bit confusing.

Page 25, Line 3: Define VRS earlier if not done already

Section 5.2, 5.3: These sections seems a bit tacked on to a very detailed earlier analysis. Is there any recommendation about how to deal with the high-NO2 data? Is there a limit at which the data is questionable? Are these cases flagged?

Technical Comments:

Abstract, Line 16: Change "∼2" to "a factor of ∼2"

Page 2, Line 14: change to "in both the troposphere and stratosphere"

Figure 3 caption: I think "d,f" should be "c,f".

---

## Author Comment (AC2) · 13 Feb 2020

**S5P/TROPOMI NO2 slant column retrieval: method, stability, uncertainties, and comparisons against OMI**

by Jos van Geffen, et al.; amt-2019-471

**Response to anonymous referee #2**

RC The manuscript "S5P/TROPOMI NO2 slant column retrieval: method, stability, uncertainties, and comparisons against OMI" provides a detailed description of the DOAS slant column fitting for TROPOMI NO2 measurements and discusses important issues related to the method, instrument, and verification. The stability and uncertainty analysis is particularly interesting. The manuscript is well organized and written. I recommend publication after a few minor revisions.

We thank the referee for the kind words and for reading the manuscript in great details.

Changes to the manuscript are based on the comments and suggestions of three referees. In addition we have extended the data record of the paper by 3 months, which has lead to updating some figures and numbers, but has *not* affected the conclusions of the paper.

In the following we answer the specific comments of referee #2.

RC P4 L7 Is there a particular reason that the VIS band stops at 496 nm? While quite a few literatures numbered the TROPOMI VIS band until 500 nm.

Due to the so-called "spectral smile" the wavelength assignment across the swath to the spectral pixels is not constant. Regarding the wavelength range of Band 4 (given in nm):

- $\circ\,$  at the edges of the swath the wavelength ranges from 397.91 for row 0 and 397.79 for row 449 to 496.01 and 495.90, respectively
- $\circ\,$  at the centre of the swath, the start wavelength is highest for rows 222 and 223: 400.29, and ranges to 498.58

In other words: only the wavelength range 400–496 is available along all detector rows.

RC P4 L20 Please give the full name of ATBD.

Done.

RC P4 S2.1.2 Please give a short description of the saturation (perhaps also blooming) effect for TROPOMI, which is mentioned and analyzed in the result section and important for TROPOMI NO2 measurements.

A few lines are added to the end of Sect. 2.1.1 [P4,L19 in the revised text]:

Over very bright radiance scenes, such as high clouds, the CCD detectors containing band 4 (Visible, e.g. used for  $NO_2$  retrieval) and band 6 (NIR, e.g. used for cloud data retrieval) may show saturation effects (Ludewig et al., 2020), leading to lower-than-expected radiances for certain spectral (i.e. wavelength) pixels. In large saturation cases, charge blooming may occur: excess charge flows from saturated into neighbouring detector (ground) pixels in the row direction, resulting in higher-thanexpected radiances for certain spectral pixels. Version 1.0.0 of the level-1b spectra contains flagging for saturation but not for blooming; version 2.0.0 will also have flagging for blooming (Ludewig et al., 2020).

RC P5 L12 What would be the implication of increasing the size towards the edge when comparing the OMI and TROPOMI measurements?

Given the difference in ground pixel size, OMI and TROPOMI measurements cannot be compared directly, not even when their orbits exactly overlap: OMI's ground pixels cover a larger geographical area than TROPOMI's, and this difference increases towards the swath edges. The only way to compare the two quantitatively is map their measurements on a common lat-lon grid, as done for Fig. 1 and 6.

RC P6 Table 1 I recommend to add the retrieval processor, namely TROPOMI (if no other name) vs QDOAS, before the date version. Please also add information of OMI/ OMNO2A (almost same as TROPOMI) in the title or as table footnote etc.

Processor names are added, as suggested. For details of the OMI/OMNO2A retrieval, a footnote refers to Sect. 3.4 (the text of which is somewhat extended, to cover both v1.2 and v2.0 of that processor), because otherwise the table would become unnecessarily complicated.

RC P9 L23 Please give rough numbers of the magnitude for OMI?

Actually the amplitude of the seasonal cycle in OMI's visible channel is comparable to TROPOMI's, as shown by Schenkeveld, et al. (2017) in their Fig. 34, as referee #1 pointed out correctly. The manuscript text has been adapted accordingly [P10,L3-5]:

A similar seasonal variation of similar amplitude is seen in the wavelength calibration data of OMI's visible channel (Schenkeveld et al., 2017, Fig. 34). Both for TROPOMI and OMI this amplitude does not exceed scatter levels and is thus well within instrument requirements.

RC P11 Eq 9 The intensity offset term shall not be placed in the same parentheses with lamda.

Oops, misplaced parenthesis in Eq. (9) – sorry: has been corrected:  $(I(\lambda) + P_{\text{off}}(\lambda) \cdot S_{\text{off}})$

RC P12 L10 Please give the full name of VCD.

You are right, this is the first time in the paper "VCD" is used – done.

RC P12 L15 What is the implication and how large is the impact of changing the LM solver to OE method with Gauss-Newton?

Test with both solvers in the wavelength calibration step, performed when setting up OE, have shown that both solvers essential give the same results and take up roughly the same cpu time. For the wavelength calibration OE has the advantage over LM that OE's solution is limited by the a-priori error value set on the wavelength shift (0.07 nm, i.e. 1/3-rd of the spectral sampling, thus guaranteeing that the shift will never be larger than the spectral sampling distance) while LM's solution is not restricted in any way. In addition adding the Ring effect in the wavelength calibration of the radiance using OE allows for the calibration to take place over a wider wavelength window than used in OMNO2A with LM. The text of Sect. 3.4 has been updated [P12,L23ff]:

... with the exception that  $\chi^2$  is minimised using a Levenberg-Marquardt (LM) solver, wavelength calibration is performed over part of the NO2 fit window (409 – 428 nm), the 2005-average irradiance spectrum as reference, and an older ozone reference spectrum (van Geffen et al., 2015). Tests have shown that the LM and OE solvers essentially give the same fit results when used with the same settings.

And in Sect. 3.2.1 [P9, L19ff]:

For the  $I(\lambda)$  calibration a 2nd order polynomial as well as a term representing the Ring effect are included: the model function used for the radiance wavelength calibration is a modified version of Eq. (2); including the Ring effect allows for a wavelength calibration to be performed across the full fit window. For the  $E_0(\lambda)$  calibration the Ring term is obviously excluded. The a-priori error of the wavelength shift is set to 0.07 nm, 1/3-rd of the spectral sampling in the NO2 wavelength range, so as to ensure that  $w_s$  will not exceed the spectral sampling distance. RC P14 L17 & P15 L20 Following the previous question, is the difference between TROPOMI and OMNO2A (Fig 3c and 3f) mainly resulted from the difference in the wavelength calibration window (I assume not the mathematic solver)?

As mentioned at the previous question, the solver is not likely to be the reason for the differences seen in the panels. The difference in the wavelength calibration window indeed has a large impact on the results; this has been added to the text [P14, L27-30]:

Differences in results of the OMNO2A and TROPOMI processor are likely mainly due to differences in the wavelength calibration: TROPOMI's radiance wavelength calibration includes a correction for the Ring effect, which allows the use of a larger calibration window (in casu the  $NO_2$  fit window; viz. Sect. 3.2.1), while OMNO2A's calibration window is necessarily limited (viz. Sect. 3.4).

Other differences between the two are minor (e.g. the sampling of the cross-sections, the precise selection of first and last wavelengths in the fit window); providing this much detail in the paper is not necessary,

RC P15 L6 Previous introduction has mentioned that TROPOMI measurements suffering from saturation are filtered out. But the residual saturation is still affecting the retrieval (even not strongly) in Fig 3d-e. Is there any recommendation of further removing the effect during retrieval or data using?

Spectral pixels flagged in the L1B spectra as saturated are removed from the fit, and in v1.2-v1.3 of the TROPOMI processor ground pixels with more than 3 such flags are not processed (the 405–465 fit window has about 305 spectral pixels).

Ground pixels with 1 to 3 flags will give more or less normal-looking SCDs but are likely to give markedly higher SCD error levels. In addition, ground pixels next to saturation cases may suffer from blooming, which possibly affects the SCDs and certainly will increase the SCD error.

As mentioned in Sect. 3.2 ground pixels with large SCD error values are flagged in the final data product as unreliable. Filtering on the SCD error was *not* done for the data in Fig. 3.

In the forthcoming v2.1 processing, the L1B flagging will be improved and the NO2 algorithm will use an outlier removal, leading to more spectral pixels being removed from the fit. As a result of this, we can allow for more spectra to enter the SCD retrieval and we can expect cases around saturation effects to give us more reliable SCDs, also for previously discarded ground pixels, albeit with somewhat elevated SCD error levels.

RC P25 L3 Please give the full name of VRS (introduced already in Sect 3).

Done.

RC P26 L13 Even the air pollution is reduced in China, it is still very possible to see columns of a few 10e16 molec/cm2 (not optically thin), particularly in Winter. With this pollution level, the boundary AMF will also show spectral features, and this nonlinearity will contribute a few percent difference to the retrieval during pollution episode. Please rewrite the statements and perhaps also provide the example for Winter for China (at the moment is July for Africa).

The example given referred to the orbit with the highest number of high NO2 values in any of the orbits from that month, July 2018, which turns out to be over Africa.

In Jan. 2019 the highest GCDs are indeed found over China (those over Africa are much lower than in July). The top 5 in highest number of ground pixels with GCD > 300 (using unit  $\mu$ mol/m2 for GCD and SCD error here for brevity) is:

○ 1609 for orbit 06580 of 20 Jan.: highest GCD is 512 ± 14; highest SCD error is 18
○ 1052 for orbit 06566 of 19 Jan.: highest GCD is 620 ± 15; highest SCD error is 15
○ 1045 for orbit 06495 of 14 Jan.: highest GCD is 581 ± 12; highest SCD error is 17
○ 577 for orbit 06637 of 24 Jan.: highest GCD is 701 ± 16; highest SCD error is 16
○ 519 for orbit 06466 of 12 Jan.: highest GCD is 549 ± 16; highest SCD error is 16
None of these values seems to be exceptionally high and reason to worry.

indeed a better example, as the discussion in this section is about high  $NO_2$  in China) [P27,L29ff]:

 $\dots$  it is currently unlikely to encounter NO2 concentrations that are not optically thin in the TROPOMI data, except in a few individual pixels.

NO2 concentration over China are highest in winter. In Jan. 2019, for example, the highest GCD found over China is  $701 \pm 16 \ \mu \text{mol/m}^2$  in orbit 06637 (24 Jan.), which has 577 pixels (0.05% of the 1204367 pixels with a successful retrieval) with a GCD exceeding 300  $\mu \text{mol/m}^2$ ; 73 pixels have a GCD values exceeding 400  $\mu \text{mol/m}^2$ . Orbit 06580 (20 Jan.) has in that month the largest number of pixels with a GCD exceeding 300  $\mu \text{mol/m}^2$ , namely 1609, with a peak value  $512 \pm 14 \ \mu \text{mol/m}^2$ ; 256 pixels have a GCD values exceeding 400  $\mu \text{mol/m}^2$ .

RC P28 L3 Do you mean case QDOAS case 4?

Corrected: it's case 3 [P29,L20]; the "case 3" mentioned a few lines down [P29,L25] has to be "case 2" (leftovers from earlier manuscript version; sorry).

---

## Author Comment (AC1)

**S5P/TROPOMI NO2 slant column retrieval: method, stability, uncertainties, and comparisons against OMI**
*by Jos van Geffen, et al.; amt-2019-471*

**Response to anonymous referee #1**

RC Any early assessment of the data quality for a major remote-sensing mission (such as TROPOMI) is always welcome, and the study in review, "S5P/TROPOMI NO2 slant column retrieval: method, stability, uncertainties, and comparisons against OMI", by van Geffen et al., delivers a timely and conscientious report on the particulars of the NO2 slant column density retrievals from the TROPOMI data. The article is well-written and reasonably compact. I perceive no major problems that may delay a prompt publication. My comments are mostly of a clarifying or editorial nature.

We thank the referee for the kind words and for reading the manuscript in great details.

Changes to the manuscript are based on the comments and suggestions of three referees. In addition we have extended the data record of the paper by 3 months, which has lead to updating some figures and numbers, but has *not* affected the conclusions of the paper.

*In the following we answer the specific comments of referee #1.*

RC P.8,l.11: The smaller-pixel size doubles the percentage of flagged NO2 retrievals. This leads to additional $\sim 2 * 10^3$ flags/orbit compared to the larger-pixel case. Please comment, if you may, on the possible cause of this doubling. Are these flagged retrievals randomly distributed across the orbit? Are they mostly related to the saturated pixels?

From the way its written it indeeds seems like the number of pixels with SCD error $> 33$ $\mu$mol/m$^2$ doubles with the coming of the smaller pixel size, compared to the total number of pixels for which an SCD error is computed. The numbers are based on comparing complete large pixel orbits from 20190403 ($0.05 \pm 0.03\%$; in absolute numbers: $627 \pm 357$) against the small pixel test orbits from 20180405 ($0.12 \pm 0.05\%$; absolute: $1872 \pm 817$).
Due to the reduces pixel size, the SCD error increases, as the paper shows, by about 10%. Hence, it would be more fair to compare the number of small pixels with SCD error $> 33$ $\mu$mol/m$^2$ to large pixels with SCD error $> 30$ $\mu$mol/m$^2$: the latter is $0.08 \pm 0.03\%$ (absolute: $1037 \pm 430$). In other words, the relative number of high SCD error pixels increases from 0.08% to 0.12% due to the pixel size reduction.
BUT here we are talking about averages over the full distribution of SCD errors, while the high SCD error values are in the tail of that distribution and the behaviour of tails of distributions may differ quite a bit from the behaviour of the average. Hence it's hard to say whether this increase from 0.08% to 0.12% is unexpected or not. In any case, it would be better to phrase this more carefully in the paper, also keeping in mind that comparing numbers from different days is not trivial due to differences in atmospheric circumstances.
In both types of orbits, the orbits passing over the South Atlantic Anomaly have the largest number of cases of larger SCD errors. Other concentrations of pixels with larger SCD errors are found along the edges of high bright clouds, where they are associated with saturation.
New text [P8,L18-20 in revised version]:

SCD error values this large occur rarely: usually $< 0.1\%$ of the pixels per orbit with original ground pixel sizes; for the smaller size pixel orbits there are about 50% more pixels with high SCD error values (based on one test day of data), taking into account that the SCD error itself increases with reduced pixel size.

RC P.9,l.16. Please comment on the evolution of the radiance-irradiance wavelength difference along the TROPOMI orbit.

The irradiance wavelength shifts in Fig. 2a are derived from the one irradiance measurement, hence that does not change along the orbit. The radiance $w_s$ in Fig. 2a is, as mentioned, an average over the 30-degree (sub-satellite) Tropical Latitude range, so as to filter out large scanline-to-scanline variation due to atmospheric circumstances: the presence of clouds may lead to large along-track variations in $w_s$ that are not related to possible instrumental issues, which is what we're looking for here.
When taking other 30-degree latitude ranges the across-track shape shown in Fig. 2a does not change visibly, while there is a very small change in the average $\overline{w_s}$ (i.e. the dotted line in Fig. 2a) from about 23 to about $24 \times 10^{-3}$ nm, going south to north. Why $\overline{w_s}$ increases from south to north is not known, but the change is very small (5% at most) and well within requirements, so we can safely ignore it further. Text added to the manuscript [P9,L26-28]:

> When taking a different latitude range the across-track shape of the radiance wavelength shift shown in Fig. 2a does not noticeably change, while the absolute value of the average shifts increases by about 5% going south to north – it is not known what causes this small increase, but it is well within instrument specifications.

RC P.9,l.23: "A similar seasonal variation, though larger in magnitude, is seen in the OMI wavelength calibration data...". Fig. 34 from Schenkeveld et al. (2017) shows comparable (early-mission OMI data) magnitudes of the seasonal $< w_s >$ variability, without any detectable long-term trends in the OMI VIS channel. Please re-phrase ll. 23-27 accordingly.

Thank you for pointing this out, you are quite right.
The seasonal amplitude is comparable – the manuscript text has been adapted, with the remark on the temperature of the optical bench removed [P10,L3-5]:

> A similar seasonal variation of similar amplitude is seen in the wavelength calibration data of OMI's visible channel (Schenkeveld et al., 2017, Fig. 34). Both for TROPOMI and OMI this amplitude does not exceed scatter levels and is thus well within instrument requirements.

RC P.11, Eq. (9) As written, the intesity offset implies a [potentially] non-linear change of the wavelength sampling for radiances. Please check.

Oops, misplaced parenthesis in Eq. (9) – sorry: has been corrected: $(I(\lambda) + P_{\text{off}}(\lambda) \cdot S_{\text{off}})$

RC P.12,l.1. "For OMI/QA4ECV both a shift and stretch are fitted;..." Is the stretch term essential for OMI? Please comment.

For by far most OMI ground pixels the stretch is $\times 10^{-4}$ or less; only close to begin and end of orbits, i.e. at high SZA, the stretch may be a little larger, up to a few $\times 10^{-3}$.

RC P.12, Sect.3.4 - Please specify whether OMNO2A v2.0 uses the intensity offset term.

It does not – added that to the text [P12,L20-21].

RC P.12, Sect.4.1, 1st par. - Please comment on how representative this particular TROPOMI orbit is. Have similar comparisons been done for different seasons? Can the shown trends be safely extrapolated on a ~year-round sample? This in

particular applies to the tests shown in Fig.3b,e. To some extent, this question is answered in P.19. It is worthwhile to make a general statement earlier on.

Note first of all that we do *not* extrapolate trends: we show daily Pacific Ocean orbits data.

The orbit used in Fig. 3 is an arbitrary choice. Obviously, the average GCD value itself varies over time due to atmospheric circumstances, but the overall characteristics of the GCD and the SCD error – the across-track shape and stripiness – are representative as is shown by the stability of the wavelength calibration shifts (Fig. 2), of the stripe amplitude (Fig. 4), and of statistical and DOAS uncertainties (Fig. 8).

Some words on the representativity are added at the beginning of Sect. 4.1 [P14,L8-9]:

> The TROPOMI orbit used here is representative for all Pacific Ocean orbits in across-track shape and variability, as is shown in subsequent sections by the stability of stripe amplitude (Sect. 4.3) and slant column uncertainties (Sect. 4.6).

RC P.14, last par. Perhaps it would be worthwhile to point to the big difference in the striping patterns delivered by various OMI algorithms, also mentioning that the forthcoming Sect. 4.3 quotes exclusively the OMI/QA4ECV stats.

Good point – text has been adapted [P14,L32ff]:

> Note that the across-track striping in the OMI results differs markedly between the different processor results, which is related to a combination of processor differences and the response to instrumental issues (OMI striping data quoted in Sect. 4.3 is taken from OMI/QA4ECV).

RC P.15,l.5. "The increased SCD error visible in the TROPOMI data of Fig. 3d-e around $\theta \approx +20$deg is related to the presence of saturation effects above bright clouds along this particular orbit." Nevertheless, most of the corresponding saturation-affected GCD values seem normal. Does this imply that the 405-465nm retrieval range retains enough of the saturation-free data to provide ∼normal-looking SCDs? Is there anything else helping to stabilize the saturation-affected retrievals? Please comment on.

Spectral pixels flagged in the L1B spectra as saturated are removed from the fit, and in v1.2-v1.3 of the TROPOMI processor ground pixels with more than 3 such flags are not processed (the 405–465 fit window has about 305 spectral pixels).

Ground pixels with 1 to 3 flags will give more or less normal-looking SCDs but are likely to give markedly higher SCD error levels. In addition, ground pixels next to saturation cases may suffer from blooming, which possibly affects the SCDs and certainly will increase the SCD error.

As mentioned in Sect. 3.2 ground pixels with large SCD error values are flagged in the final data product as unreliable. Filtering on the SCD error was *not* done for the data in Fig. 3.

In the forthcoming v2.1 processing, the L1B flagging will be improved and the NO2 algorithm will use an outlier removal, leading to more spectral pixels being removed from the fit. As a result of this, we can allow for more spectra to enter the SCD retrieval and we can expect cases around saturation effects to give us more reliable SCDs, also for previously discarded ground pixels, albeit with somewhat elevated SCD error levels.

RC P.16,l.18. The 4-times difference in the QDOAS/TROPOMI RMS estimates is hard to overlook, indeed. Considering the typical S/N∼1500 in the TROPOMI data, one may side with the RMS∼$8 * 10^{-4}$ provided by QDOAS. Similar- magnitude

RMS$\sim 0.5 - 1.0 * 10^{-4}$ is frequently quoted by various groups working with $\sim$similar S/N data sets from various spacecrafts. The cited (Table 2) RMS$\sim 0.2 * 10^{-4}$ seems like an overly optimistic assessment. Any comment?

The point behind this is that the RMS error definitions in the IF and ODF modes differ. QDOAS ODF uses Eq. (7) and TROPOMI IF used Eq. (4); the relation between these two is discussed in Appendix B. QDOAS IF gives a 9% higher RMS error that QDOAS ODF (mentioned in the preceeding "dash" point), hence that mode seems to be using an RMS error definition different from both Eq. (4) and Eq. (7).

The manuscript is improved by rephrasing these two "dash" points [P17,L1-5]:

– The RMS error calculation of the TROPOMI IF mode and the QDOAS ODF mode, given in Eq. (4) and Eq. (8), respectively, lead to different results; a relation between these two is given in App. B.

– Given that the RMS error in the QDOAS IF mode is $\sim 9\%$ higher than in the QDOAS ODF mode, indicates that the RMS definitions of these two QDOAS modes may be slightly different for the two modes, and that the definition of the QDOAS IF mode is different from the TROPOMI IF mode.

RC P.18,l.19 "Fig. 4c shows this RMS...". Please provide more details on how this value was calculated.

Actually there is not more to it than just that: the RMS of the SCD stripe amplitude, i.e. the blue line in Fig. 4a. The equation has been added [P19,L3-4]:

A measure for the stability of the SCD stripe amplitude is the RMS of the across-track stripe amplitude, i.e. of the blue line in Fig. 4a: $\sqrt{\{\overline{\sum_i (N_{s,i}^{str})^2}\}}$, with summation over rows $i = 0, 1, \ldots, 449$.

RC P.19,l.9 Is de-striping applied both to TROPOMI and OMI? Please clarify in the text.

Yes, and added [P19,L17]:

... after applying the respective de-striping of the SCDs described in the previous subsection on both datasets.

RC Fig.6. Since the intensity offset has the highest impact over the cloud-free, clear-water ocean areas (Sect. 5), one may test this by segregating the data in Fig.6a into two cases (cloudy and cloud-free; probably, selecting even more extreme cases than in Fig.8) and commenting on. I consider this as an important test, in light of the findings from Oldeman (2018).

Fig. 6 only shows cloud-free cases (cloud radiance fraction $< 0.5$), but is indeed a good idea to add cloudy cases – that has been done in the from of a new figure, Fig. 7, and the text has been extended accordingly [P16,L18ff]:

Fig. 6 shows the scatter plot of the TROPOMI and OMI/Q4ERCV GCDs of (almost) clear-sky ground pixels (i.e. cloud radiance fraction $< 0.5$) for July 2018 for two regions: the remote Pacific Ocean and the polluted area covering India and China on the Northern Hemisphere; the definition of these two areas is included in the figure panel legends. Both regions show a very good correlation with $R^2 \approx 0.99$. Over the Pacific Ocean area (Fig. 6a) the clear-sky TROPOMI GCD is on average $2.20 \pm 1.65$ $\mu$mol/m$^2$ ($1.33 \pm 0.99 \times 10^{14}$ molec/cm$^2$) or $5.23 \pm 3.93\%$ larger than the OMI/QA4ECV GCD. For Jan. 2019 the result (not shown) is quite similar: the clear-sky TROPOMI GCD over the Pacific Ocean is on average $2.19 \pm 1.56$ $\mu$mol/m$^2$ or $5.78 \pm 4.61\%$ larger than OMI/QA4ECV. Over the polluted India-to-China area (Fig. 6b) the clear-sky TROPOMI GCD is on average $2.02 \pm 2.08$ $\mu$mol/m$^2$ or $3.79 \pm 4.06\%$ larger than OMI/QA4ECV, i.e. the relative difference is a little smaller than from the Pacific Ocean.

For cloudy pixels (i.e. cloud radiance fraction $> 0.5$) the difference between the

> TROPOMI and OMI/QA4ECV GCD is smaller, both in absolute and in relative terms, and the scatter is less, as can be seen from Fig. 7. Over the Pacific Ocean area (Fig. 7a) the cloudy TROPOMI GCD is on average $1.27\pm0.93$ $\mu$mol/m$^2$ ($0.76\pm0.56\times10^{14}$ molec/cm$^2$) or $3.04\pm2.39\%$ larger than the OMI/QA4ECV GCD. Over the polluted India-to-China area (Fig. 7b) the clear-sky TROPOMI GCD is on average $1.38\pm1.26$ $\mu$mol/m$^2$ or $2.74\pm2.37\%$ larger than OMI/QA4ECV.

Over land (Fig. 6b and Fig 7b) the differences are less than over ocean (Fig. 6a and Fig 7a), and over clouds (Fig. 7) the differences are less than over clear-sky cases (Fig. 6). Since clear-water cases show larger differences than land and cloudy cases, it thus seems that the intensity offset correction may be correction some absorption effect in open waters but not only such effects. The discussion in Sect. 5.1 has been extended in this sense [P26,L22ff]:

> The quantitative comparison discussed in Sect. 4.4 revealed that for clear-sky cases (Fig. 6) the differences are a little larger than for the cloudy cases (Fig. 7), and for clear-sky cases the difference is larger for the remote Pacific Ocean area (almost completely water) than for the polluted India-to-China area (mainly land surface), while for the cloudy cases the differences are comparable for the two areas. These differences thus seems to indicate that the IOC may be correcting for some absorption effects in ocean waters, but not only for such absorption effects given that the reduction in GCD is also seen over land and over clouds.

RC P.22,l.2 "The fact that the GCD value itself (Fig. 7c) is not appreciably affected by the time difference is very reassuring..." Actually, I find this really puzzling: cf. Fig.7b and 7c. The SCD error, as anticipated, linearly increases in time in Fig.7b, but remains essentially flat in Fig.7c. Please help me (and the readers) to interpret this.

Fig. 7c does (in the revised version it's Fig. 8) *not* show the SCD error, I'm afraid. Instead:
  ○ the SCD error and its stripiness (stddev) are given in Fig. 7b,
  ○ the GCD and its stripines (stddev) are given in Fig. 7c.

Fig. 7c shows that the GCD itself (red) remains the same – which is the reassuring point made – while at the same time the stripiness increases with increasing time difference between radiance and irradiance.

The confusion may have come from the fact that in the Fig. 7b and Fig. 7c *red and blue solid* lines are used for one case of SCD error or GCD and its stddev, while *red and blue dashed* lines are used for the other case. This is a bit counter-intuitive as one would expect the quantity to be given by a solid line and the stddev by a dashed line. But in that case the solid lines for the quantities almost overlap in Fig. 7b and fully overlap in Fig. 7c – that's the background for the original choice.

But your comment and the comment of referee #3 has shown that the choice made is too confusing, hence the more intuitive approach is used now, with an updated figure caption [P23].

RC P.23,l.12 Table 3: Please specify whether the TROPOMI (presumably, yes) and OMI (?) SCDs were de-striped.

No, the SCD data is *not* de-striped for the statistical uncertainty analysis.

De-striping would introduce model information in the SCD data and we want to investigate the uncertainty if the SCD data itself, i.e. coming directly from the instrument. Note further that the DOAS uncertainty is linked to the not de-striped SCDs, i.e. if de-striped SCDs were used for the statistical uncertainty, the link between the quantities would be weaker.

RC  P.23,l.28 "From Fig. 8 it is furthermore clear that the statistical and the DOAS
uncertainties of TROPOMI appear to be stable over the currently available data
period." This is not what Fig.8 shows (the Aug. 6 jump). Please re-phrase.

No – the Fig. 8 does show that the instrument is stable over time.
The jump at 7 Aug. is *not* an instrumental issue/problem, but a configuration choice.
This is also indicated by the fact that the standard deviation of the quantities given in
Table 3 are almost the same before and after the pixel size change. New text [P26,L1-5]:

> From Fig. 9 and Table 3 it is furthermore clear that the statistical and the DOAS
> uncertainties of TROPOMI appear to be stable over the currently available data
> period: the standard deviation of the quantities given in Table 3 are small and Fig. 9
> shows no systematic change over time. The jumps in the quantities on 6 Aug. 2019
> are caused by the along-track pixel size change, not by an instrumental issue, and
> this change has not affected the stability: the standard deviations of the quantities
> given in Table 3 are not markedly different between the two measurement modes.

RC  P.25,l.14 "...also because instrumental effects such as straylight and dark current are
adequately corrected for in the spectral calibration in the level 0-to-1bprocessor."
Either remove the statement or provide the references that address the subject and
are based on assessment of the post-launch data (both radiances and irradiances).

Word "adaquately" removed (stating the spectra are calibrated suffices) and two refer-
ences added: (Kleipool et al, 2018; Ludewig et al., 2020); the latter is a paper in review
for the same AMT TROPOMI Special Issue since early Feb. 2020.

RC  Section 5.2 I have a problem linking the statement "Since NO2 over the Pacific
Ocean is primarily stratospheric NO2, validation of stratospheric NO2 essentially
is also validation of Pacific Ocean NO2 SCDs." Besides the point that it should
be re-phrased, I do not see any connections between the discussed TROPOMI
retrievals over Pacific and the ZSL-DOAS/SAOZ S5PVT network that completely
avoids the Pacific basin. If the authors employed somewhat different approach than
in Lambert et al. (2019), then they should provide a detailed description of the
validation process. If this Section provides a summary of the Lambert et al. report
and nothing besides, it should say so.

The section argues that the NO2 over the Pacific Ocean is primarily located in the
stratosphere, in view of the absence of (anthropogenic) sources of NO2 in the region.
Hence, validation of stratospheric NO2 can also be seen as validation of Pacific Ocean
SCDs. The sentence quoted by the referee has been reformulated somewhat [P27,L4-5]:

> Since $NO_2$ over the Pacific Ocean, i.e. away from anthropogenic sources of $NO_2$, is
> primarily located in the stratosphere, validation of stratospheric $NO_2$ can also be
> seen as validation of Pacific Ocean $NO_2$ SCDs.

Validation of stratospheric NO2 is described by Lambert et al. (2019) and we quote
from that. We have performed no additional validation.

RC  P.26,l.7 "...and the temperature dependence of the NO2 reference spectrum (usually
corrected for a-posteriori in the AMF application) may show spectral structures..."
Do you imply: "...and the temperature correction of the NO2 VCDs (usually intro-
duced aposteriori in the AMF application) may result in spectral artifacts in the
fitting residuals that are linked to the temperature dependence of the NO2 refer-
ence spectrum." ? The NO2 reference spectrum always shows spectral structures
in the 405-465nm range...

The formulation is indeed not clear. It was formulated by Richter et al. (2014) in the
poster abstract a "the spectral signature of the temperature dependence of the NO2

absorption cross-section could be detected." The point thus is probably that the temperature dependency depends on the wavelength. Hence "may show spectral structures" is replaced by "may be wavelength dependent" [P27,L25].

RC P.27,l.7 "Since NO2 over the Pacific Ocean is primarily stratospheric NO2, validation of stratospheric NO2essentially is also vali-dation of Pacific Ocean NO2SCDs. As reported by Lambert et al. (2019..." Again, I fail to relate the discussed Pacific Ocean retrievals to the Lambert et al. report.

See above.

RC P.28,l.3 There is no QDOAS Case 6 in Table 2. Please correct.

Corrected: it's case 3 [P29,L20]; the "case 3" mentioned a few lines down [P29,L25] has to be "case 2" (leftovers from earlier manuscript version; sorry).

RC P.28,l.23 "After removal of such outliers..." Please specify how the removal is done: are the pixels corrected and re-used or completely removed from the fit?

Spectral pixels with outliers are completely removed from the fit, just as spectral pixels suffering from (too much) saturation or flagged as "bad" in the level-1b spectra. The text has been adjusted somewhat [P30,L9-10]:

> Spectral pixels with such outliers are removed completely from the measured reflectance and the DOAS fit is redone to provide ...

RC P.28,l.25 If this concerns the total, spatial x spectral number (I presume, though, that the authors speak of the spatial domain only – please clarify!), ∼5% is, actually, a sizable population of the pixels. It may grow in time, eventually leading to much noisier and probably biased (depending on the preferential location of the spikes) retrievals. Any comments on this?

We find in ∼5% of the ground pixel spectra one or more spectral pixel with an outlier (some 80,000 of the about 1.5 millioun ground pixels per orbit). This is indeed a sizeable portion, but since most of these outliers are related to either the South Atlantic Anomaly transients or saturation effects over clouds, this does not seem to be much of a problem nor an issue that is likely to grow over time. See also next point.

RC P.28,l.26 "...most of which have less than 5 outliers per ground pixel..." Please clarify that these 5 outliers happen in the spectral domain.

Indeed, 5 spectral pixels showing outliers; text adapted, along with the changes from the preceeding point [P30,L11ff]:

> Outliers occur only in a small fraction of the ground pixels: usually ∼ 5% of the successfully processed ground pixels show one or more outliers in their spectrum, and most of these ground pixels with outliers have less than 5 spectral pixels showing outliers per ground pixel; the largest effects ...

*In the following we answer the technical comments of referee #1.*

RC P.4,l.23: "...away from anthropogenic sources of NO2..."

Done.

RC P.9,l.28: "The dominant term in the overall magnitude of the radiance is the inhomogeneous illumination..." This needs to be clarified: "For a given field-of-view (row), the dominant term in the overall magnitude..." - is this what the authors meant?

Indeed, for a given field-of-view (ground pixel, not just row). The presence of clouds will likely cause differences in $w_s$ along-track, across-track and day-to-day. The text has been adapted as suggested [P10,L6].

RC  P.9, l.29 "The magnitude of the day-to-day variation in the average is much smaller than this long-term oscillation..." By 'this long-term oscillation' do you mean ...the seasonal (Fig.2b) oscillation...?

We indeed mean the oscillation in Fig. 2b; that has been added to the text [P10,L9].

RC  P.15,l.19 "...the TROPOMI processor reports 10.2% ..."

Done.

RC  P.15,l.27 "TROPOMI level-1b version 1.0.0 spectra..."

Done.

RC  P.16,l.6 "...with other configuration settings as much as possible matching those of the TROPOMI processor..."

Done.

RC  P.16,l.16 "...that the RMS definition may be different..."

Done. Thanks for reading the manuscript so carefully.

RC  P.16,l.20 "As a reference..."

Done

RC  P.18,l.4 "... is smooth rather than stripe-like over the non-contaminated areas..."

Good idea to add that.

RC  P.25,l.4 "...the intensity offset corrections are..."

Done.

RC  P.25,l.14 adequately

Done.

---

## Author Comment (AC3)

**S5P/TROPOMI NO2 slant column retrieval: method, stability, uncertainties, and comparisons against OMI**
*by Jos van Geffen, et al.; amt-2019-471*

**Response to anonymous referee #3**

RC  This paper describes the slant column density retrieval for the TROPOMI satellite instrument and gives a detailed assessment of uncertainties and comparisons with OMI. The paper is well written and despite the topic being very technical, I found the paper very clear and easy to follow. The details will be of interest to a limited set of scientists (retrieval algorithm developers mostly) but it is a thorough record of the uncertainties and preliminary temporal behaviour of the TROPOMI slant column retrievals. I recommend it be published following a few minor revisions.

We thank the referee for the kind words and for reading the manuscript in great details.

Changes to the manuscript are based on the comments and suggestions of three referees. In addition we have extended the data record of the paper by 3 months, which has lead to updating some figures and numbers, but has *not* affected the conclusions of the paper.

*In the following we answer the general and specific comments of referee #3.*

RC  The paper focussed almost entirely on an analysis of slant columns from tropical observations over the clean remote Pacific. I would have liked to have seen some slant column retrieval comparisons over an entire orbit, as some of the differences between retrieval approaches or instruments might be exaggerated at the highest solar zenith angles. Perhaps adding just a sentence or two describing how well the conclusions drawn about uncertainties, retrieval algorithm comparisons etc extend to cases other than the remote Pacific could be useful.

Pacific Ocean orbits are used for the stability and accuracy analysis of the SCD retrieval because we may assume that those orbits do not show significant (anthropogenic) tropospheric $NO_2$ concentrations.
  - The stripe amplitude stability in Sect. 4.3 uses a 30-degree tropical latitude range over the Pacific Ocean, as that is range is used to define the stripe amplitude.
  - The quantitative comparison in Sect. 4.4 uses data from multipile orbits (a) over the Pacific in the latitude range [-60:+60] and mentions results for both July 2018 and Jan. 2019, and thus covers a wide range of SZAs, and (b) over the India-to-China ares in a latitude range of [0:+60] for July 2018 only, thus also covering a fair range of SZAs as well as pollution sources.
  - The SCD uncertainty analysis in Sect. 4.6 uses Pacific Ocean orbit data in the latitude range [-60:+60] for all days of the year and thus includes a wide range of SZAs.

In short, we feel that the points raised by the referee are already well covered in the paper.

RC  Page 3, Line 11: Give units of conversion factor.

That would be something like "molecules per mol", but since "molecules" is not a proper unit, the conversion factor has the same unit at Avogadros number: 1/mol – added this unit.

RC  Page 5, Line 31: I'm a bit confused by the wording describing a satellite latitude range. How is this changing between these two end points of 1 Jan and 1 July?

With "satellite latitude range" we mean the latitude of the sub-satellite point, i.e. the data in the `satellite_latitude` variable, which approximately corresponds to the nadir viewing detector rows. The wording has been adapted to describe this more clearly [P6,L4ff]:

> To investigate the stability and uncertainties of the $NO_2$ SCD retrieval the "Tropical Latitude" (TL hereafter) range is defined as all scanlines that have their sub-satellite latitude point – corresponding approximately to the nadir viewing detector rows – within a 30° range that moves along with the seasons, in an attempt to filter out ...

RC Page 9, Line 25: Comparing to OMI but no OMI results shown, so could you give a number indicating the magnitude of OMI variations?

Actually the amplitude of the seasonal cycle in OMI's visible channel is comparable to TROPOMI's, as shown by Schenkeveld, et al. (2017) in their Fig. 34, as referee #1 pointed out correctly. The manuscript text has been adapted accordingly [P10,L3-5]:

> A similar seasonal variation of similar amplitude is seen in the wavelength calibration data of OMI's visible channel (Schenkeveld et al., 2017, Fig. 34). Both for TROPOMI and OMI this amplitude does not exceed scatter levels and is thus well within instrument requirements.

RC Page 19, Line 12: define "India and China" latitude/longitude region

Both regions are defined in the legends in the figure panels; a reference to that is included in the text [P19, L20].

RC Page 21, Line 2: degradation of 1-2% relative to what? Is this degradation in throughput per year?

Degradation of the absolute irradiance, w.r.t. the beginning of the mission; the potentially confusing word "notably" has been removed.

RC Figure 7: I find the colors of b and c very hard to follow in my mind. I think it's more common to be looking at a solid line that represents the average and a dotted line of the same color that represents a standard deviation or similar. Here they are different colors but the same pattern for a single orbit (backwards to what I'm used to). Not a significant issue but I just find it a bit confusing.

Your comment and the comment of referee #1 has shown that the choice made for the linetype is too confusing, hence the more intuitive approach is used now, with an updated figure caption, noting that the solid lines for the quantities themselfs almost overlap in Fig. 7b and fully overlap in Fig. 7c.
Note that in the revised version the figure now has number 8.

RC Page 25, Line 3: Define VRS earlier if not done already

Done.

RC Section 5.2, 5.3: These sections seems a bit tacked on to a very detailed earlier analysis. Is there any recommendation about how to deal with the high-NO2 data? Is there a limit at which the data is questionable? Are these cases flagged?

Neither section is based on more detailed analysis by the authors: Sect. 5.2 uses the S5PVT validation results to make a statement about the Pacific Ocean $NO_2$ SCDs. Sect. 5.3 discusses earlier findings regaring high $NO_2$ concentrations published by Richter et al. (2014).
As mentioned in the paper, it is unlikely that TROPOMI will detect concentrations of $NO_2$ so high that the reported concentrations are really wrong. One may in individual

ground pixels find such high values, but there are likely too few points on which to base any sensible statement.

Ground pixels with high $NO_2$ concentrations are not flagged as such; usually these concentrations will come with a somewhat elevated SCD error but not exceptionally large SCD errors (as in the example mentioned: $883 \pm 16$ $\mu$mol/m$^2$). Given that there is not a clear-cut limit between good and bad high $NO_2$ concentrations, there is no sensible criterion to yes of no flag such data.

*In the following we answer the technical comments of referee #1.*

RC Abstract, Line 16: Change "$\sim$2" to "a factor of $\sim$2"

Done.

RC Page 2, Line 14: change to "in both the troposphere and stratosphere"

Done.

RC Figure 3 caption: I think "d,f" should be "c,f".

You are right, thanks for noting this.